# Neoadjuvant antiangiogenic therapy reveals contrasts in primary and metastatic tumor efficacy

John ML Ebos[1,*], Michalis Mastri[1], Christina R Lee[2], Amanda Tracz[1], John M Hudson[2], Kristopher Attwood[3], William R Cruz-Munoz[2], Christopher Jedeszko[2], Peter Burns[2,4] & Robert S Kerbel[2,4]

## Abstract

Thousands of cancer patients are currently in clinical trials evaluating antiangiogenic therapy in the neoadjuvant setting, which is the treatment of localized primary tumors prior to surgical intervention. The rationale is that shrinking a tumor will improve surgical outcomes and minimize growth of occult micrometastatic disease—thus delaying post-surgical recurrence and improving survival. But approved VEGF pathway inhibitors have not been tested in clinically relevant neoadjuvant models that compare pre- and post-surgical treatment effects. Using mouse models of breast, kidney, and melanoma metastasis, we demonstrate that primary tumor responses to neoadjuvant VEGFR TKI treatment do not consistently correlate with improved post-surgical survival, with survival worsened in certain settings. Similar negative effects did not extend to protein-based VEGF pathway inhibitors and could be reversed with altered dose, surgical timing, and treatment duration, or when VEGFR TKIs are combined with metronomic 'anti-metastatic' chemotherapy regimens. These studies represent the first attempt to recapitulate the complex clinical parameters of neoadjuvant therapy in mice and identify a novel tool to compare systemic antiangiogenic treatment effects on localized and disseminated disease.

**Keywords** antibodies; neoadjuvant; surgery; tyrosine kinase inhibitors; VEGF
**Subject Categories** Cancer; Vascular Biology & Angiogenesis

See also: **D Biziato & M De Palma** (December 2014)

## Introduction

Eight inhibitors that block the vascular endothelial growth factor (VEGF) pathway have now been approved as first- or second-line treatment in twelve different late-stage cancer types, thus validating antiangiogenesis as a therapeutic modality in treating established metastatic disease and late-stage glioblastoma (Jayson *et al*, 2012). Stemming from these approvals, several hundred phase II and III trials were initiated to evaluate VEGF pathway inhibitors in earlier stage disease, that is, neoadjuvant (pre-surgical) and adjuvant (post-surgical) treatment settings (Ebos & Kerbel, 2011). Such 'perioperative' treatments are unique in that they typically have defined treatment durations (unlike in late-stage or advanced disease, where treatments are variable depending on response) and are guided by the hypothesis that drug efficacy in advanced metastatic disease would elicit equal or greater improvements in the earlier stages (Tanvetyanon *et al*, 2005). These benefits—shown with radiation and chemotherapy (Van Cutsem *et al*, 2009)—would theoretically include control of localized primary cancers which, in turn, would prevent occult micrometastatic disease and improve progression-free survival (PFS) (Ebos & Kerbel, 2011). However, based on recent clinical and preclinical observations, there is growing concern that VEGF pathway inhibitors may not be effective in this setting (Ebos & Kerbel, 2011). First, there have been five failed phase III adjuvant trials with VEGF pathway inhibitors, including four with the VEGF neutralizing antibody bevacizumab (in combination with chemotherapy or an anti-HER2 antibody) in colorectal carcinoma (CRC) (AVANT and C-08) (de Gramont *et al*, 2012) and triple-negative and HER2[+] breast carcinoma (BEATRICE and BETH, respectively) (Cameron *et al*, 2013), and one with the VEGF receptor tyrosine kinase inhibitor (RTKI) sorafenib in hepatocellular carcinoma (HCC) (Bruix *et al*, 2014). Second, growing preclinical evidence suggests that unexpected collateral consequences of angiogenesis inhibition may limit efficacy in preventing growth of micrometastatic lesions (Mountzios *et al*, 2014). Indeed, we and others have demonstrated that VEGF pathway inhibitors can elicit both tumor- and host-mediated reactions to therapy that can offset (reduce) benefits, or even facilitate, early-stage metastatic disease in certain instances (Ebos *et al*, 2009; Paez-Ribes *et al*, 2009). Though these latter results have thus far not been confirmed clinically in patients with advanced metastatic disease when therapy is removed (Miles *et al*, 2010; Blagoev *et al*, 2013), they underscore a gap in our current understanding of how antiangiogenic therapy may work in different disease stages. They

1   Genitourinary Section, Department of Medicine, Roswell Park Cancer Institute, Buffalo, NY, USA
2   Biological Sciences Platform, Sunnybrook Research Institute, Toronto, ON, Canada
3   Department of Biostatistics and Bioinformatics, Roswell Park Cancer Institute, Buffalo, NY, USA
4   Department of Medical Biophysics, University of Toronto, Toronto, ON, Canada
    *Corresponding author. Tel: +1 716 8454464; E-mail: John.Ebos@roswellpark.org

also raise questions about the translational value of preclinical studies in predicting clinical outcomes. This is of immediate concern as few preclinical studies have tested VEGF pathway inhibitors in clinically appropriate models of late-stage metastatic disease (Guerin *et al*, 2013), and even fewer still have modeled treatments in the perioperative setting with spontaneous metastatic disease similar to patients. For this reason, there is an urgent need to develop predictive preclinical models to evaluate the efficacy of different VEGF pathway inhibitors in localized versus micrometastatic disease.

Neoadjuvant therapy may offer significant value in this regard (de John, 2012). Two recent phase III trials examining bevacizumab (with chemotherapy) in the neoadjuvant setting demonstrated improved pathological complete response (pCR) (Bear *et al*, 2012; von Minckwitz *et al*, 2012a), and there are numerous neoadjuvant trials underway or completed in renal cell carcinoma (RCC) with VEGFR TKIs such as sunitinib (NCT00849186), axitinib (NCT01263769) and pazopanib (NCT01512186) (Bex & Haanen, 2014). The rationale behind such trials is based on several presumed/theoretical advantages of antiangiogenic therapy in the neoadjuvant setting. These include (i) primary tumor debulking to improve surgical margins and spare tissue or organs (such as nephron sparing in RCC), (ii) to assess treatment efficacy for potential use in post-surgical recurrent disease, and (iii) to prevent occult metastatic lesions not detectable at time of surgery (van der Veldt *et al*, 2008; Silberstein *et al*, 2010; Ebos & Kerbel, 2011; Fumagalli *et al*, 2012; Schott & Hayes, 2012; Bex & Haanen, 2014). Surprisingly, few preclinical studies have examined pre-surgical therapy (Padera *et al*, 2008; de Souza *et al*, 2012), and none have established appropriate parameters in preclinical models of spontaneous metastatic disease to compare the effects of neoadjuvant antiangiogenic treatment. Such studies could serve as a predictive tool to compare pre-surgical primary tumor responses to systemic therapy to post-surgical benefits, such as delayed metastatic disease and improved survival.

Using established models of spontaneous metastasis following surgical removal of orthotopically grown tumors in mice (Francia *et al*, 2011), we have developed a methodical approach to evaluate neoadjuvant therapy and assess the value of primary tumor responses as predictors of eventual (post-surgical) metastatic recurrence. Our results show that primary tumor responses and post-surgical metastatic recurrence rates after VEGFR TKI treatment do not consistently correlate, and reveal the potential that primary tumor reduction can be offset by worsened post-surgical survival. Importantly, such effects could be minimized with altered dose, surgical timing, and treatment durations, as well as the addition of metronomic chemotherapy regimens. Interestingly, protein-based VEGF pathway inhibitors (including VEGF and VEGFR-2 inhibitors) provide an example of how drug efficacies can differ within drug classes. Taken together, our models help to distinguish therapeutic efficacy as 'anti-primary' and 'anti-metastatic' (or both), could help explain some recent high-profile trial failures, and may serve to predict outcomes for patients currently receiving neoadjuvant antiangiogenic therapy.

# Results

## Optimal neoadjuvant treatment and surgical parameters differ in multiple metastatic models

To evaluate neoadjuvant therapy in mice, we first defined an optimal window for neoadjuvant therapy using four tumor models of spontaneous metastasis that involved orthotopic implantation of tumor cells followed by primary tumor resection. Human tumor xenograft models included breast (LM2-4$^{LUC+}$) (Ebos *et al*, 2008), melanoma (WM113/6-4L) (Cruz-Munoz *et al*, 2008), and kidney (SN12-PM6$^{LUC+}$) cell lines in SCID mice, while a mouse syngenic model utilized the kidney cell line (RENCA$^{LUC+}$) (Tracz *et al*, 2014) in BALB/c. Optimization of the models to evaluate neoadjuvant therapy examined three parameters: (i) determination of metastatic potential (MP), used to identify the tumor size prior to surgery necessary to ensure sufficient metastatic disease; (ii) optimal surgical time (OST), used to define a tumor growth period sufficient to elicit spontaneous metastasis; and (iii) residual cancer burden (RCB), used to allow for potential comparisons with clinical parameters of pCR, monitoring of surgical variability as well as exclusion of mice with obvious non-localized disease at surgery (detailed in Supplementary Results and Supplementary Fig S1A–H).

## Primary tumor responses following neoadjuvant sunitinib treatment do not correlate with post-surgical survival in metastatic kidney and melanoma models

Short-term neoadjuvant sunitinib treatments were compared in three models of varying response to therapy in the pre-surgical setting. In the first, SN12-PM6$^{LUC+}$ cells were implanted into the subcapsular space of kidneys in SCID mice and randomized

---

**Figure 1. Primary tumor response to neoadjuvant sunitinib treatment is not predictive of post-surgical survival in multiple models of metastasis.**

A   BLI of SCID mice bearing orthotopic human SN12-PM6$^{LUC+}$ renal tumors receiving neoadjuvant sunitinib for 14 days.
B   BLI of mice before and after nephrectomy (SN12-PM6$^{LUC+}$ model).
C   Corresponding quantification of resected kidney BLI (left panel) and kidney weight (right panel) following neoadjuvant sunitinib treatment cessation (SN12-PM6$^{LUC+}$ model).
D   Post-surgical survival (SN12-PM6$^{LUC+}$ model).
E   BLI of BALB/c mice bearing orthotopic mouse RENCA$^{LUC+}$ renal tumors receiving neoadjuvant sunitinib for 14 days.
F   BLI of mice before and after nephrectomy (RENCA$^{LUC+}$ model).
G   Corresponding quantification of resected kidney BLI (left panel) and kidney weight (right panel) following neoadjuvant sunitinib treatment cessation (RENCA$^{LUC+}$ model).
H   Post-surgical survival (RENCA$^{LUC+}$ model).
I   Heatmap summary of metastatic distribution by visual scoring at individual mouse endpoint following neoadjuvant sunitinib treatment and tumor resection (SN12-PM6$^{LUC+}$, RENCA$^{LUC+}$, and human WM113/6-4L melanoma tumor model).

Data information: Symbols and bars for box and whiskers plot: median (line), upper/lower quartile (box), min/max (error bars). Survival analysis: hazard ratio (HR), confidence interval (CI), overall survival (OS) based on Kaplan–Meier or Cox regression analysis. *N* = 8–12 mice per group. BLI, bioluminescence imaging; Neoadj. Tx, neoadjuvant treatment; *P* < 0.05, **P* < 0.01 compared to control.

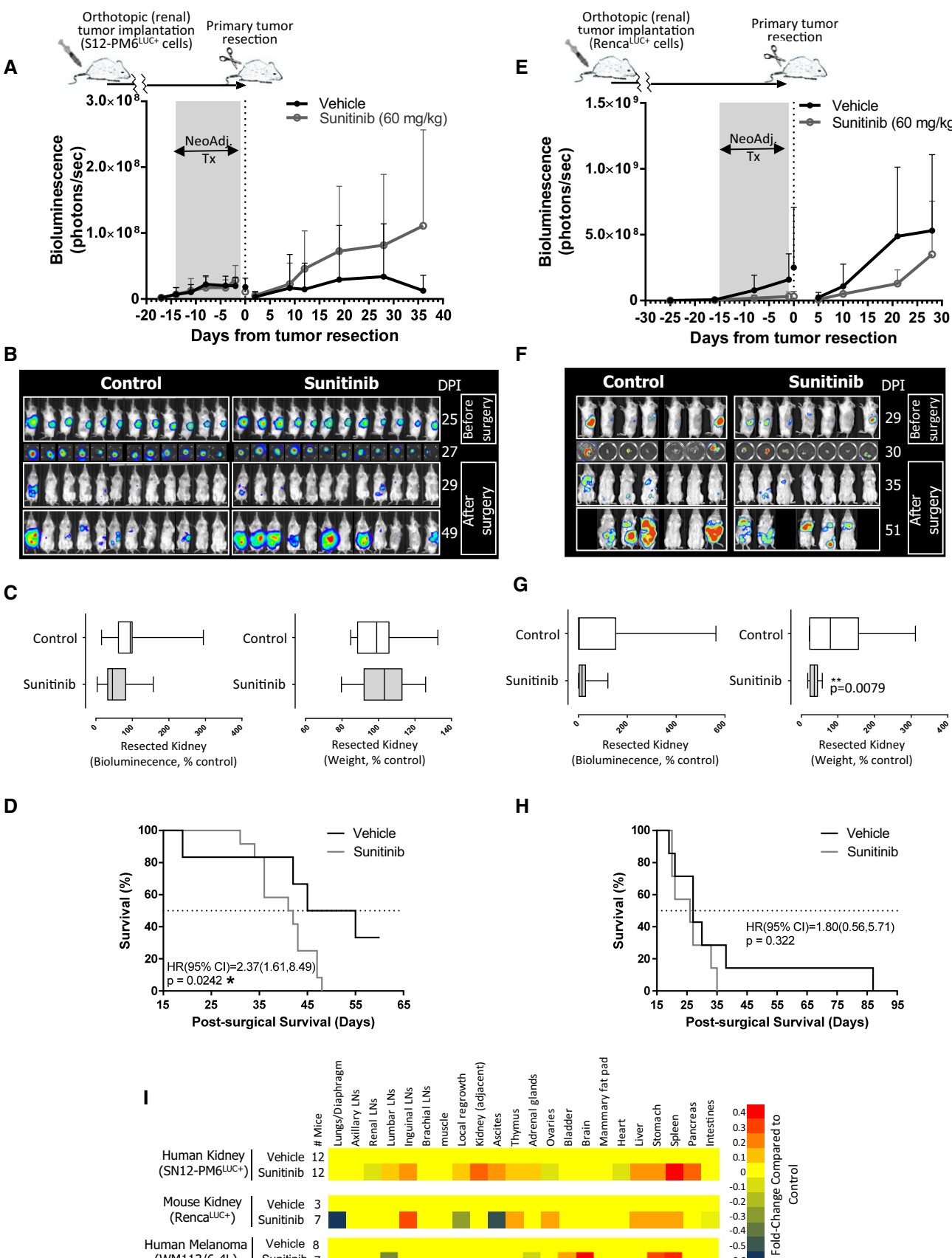

**Figure 1.**

into groups of corresponding size based on bioluminescence (see Materials and Methods for details). Neoadjuvant sunitinib treatment (60 mg/kg/day for 14 days) yielded no reductions in overall bioluminescence (BLI) (Fig 1A and B), or overall kidney weight or kidney BLI following surgical removal, compared to vehicle controls (Fig 1C, left and right panel, respectively). However, upon treatment cessation and surgical removal of the kidney, sunitinib-treated mice had significantly decreased overall survival (Fig 1D). In a second model, BALB/c mice bearing orthotopic RENCA$^{LUC+}$ tumors received neoadjuvant treatment (60 mg/kg sunitinib for 14 days) yielding significantly reduced pre-surgical BLI (Fig 1E and F) and reduced resected kidney BLI and weights (see Fig 1G, left and right panel, respectively). However, these significant pre-surgical benefits did not lead to improvements in post-surgical survival (Fig 1H). A third model yielded similar differences in pre- and post-surgical comparisons with vehicle-treated controls. SCID mice bearing orthotopic melanoma (WM113/6-4L) cells treated with neoadjuvant sunitinib (60 mg/kg for 14 days) led to a trend of reduced tumor size and weight following surgical resection but yielded a trend of worsened post-surgical survival (Supplementary Fig S2A–C, both did not reach statistical significance). Metastatic sites at the endpoint were visually assessed in all three models and compared to respective vehicle controls to test whether neoadjuvant treatment influenced post-surgical metastatic disease progression patterns. Stomachs and spleens had consistent increases in metastasis compared to controls in response to treatment in all models, but no clear trends suggested therapy-induced progression pattern differences (Fig 1I). Together, all three models showed that pre-surgical effects of neoadjuvant treatment did not predict for similar effects in the post-surgical setting with benefits (or non-benefits) leading to consistently worsened outcomes (i.e., no benefit or decreased survival).

### Dose, treatment duration, and surgical timing can improve neoadjuvant sunitinib treatment efficacy outcomes in a metastatic breast model

We next undertook experiments to determine whether treatment dose, duration, and surgical resection timing had the potential to impact post-surgical outcomes following pre-surgical neoadjuvant therapy (Fumagalli *et al*, 2012). Following implantation of human LM2-4$^{LUC+}$ breast cells into the mammary fat pads of SCID mice, sunitinib (60 mg/kg) was administered daily for 14 days prior to surgical removal of the primary tumor. Separately, the chemotherapeutic drug cyclophosphamide (CTX) was administered at the maximum tolerated dose (MTD) for the same period (Schott & Hayes,

2012). Both sunitinib and CTX MTD led to significant reductions in primary tumor volume (Fig 2A) and reduced excised tumor weight (Fig 2B) compared to vehicle-treated controls. CTX MTD neoadjuvant treatment led to a delay in post-surgical metastatic recurrence and improved survival; however, a similar benefit was not seen following neoadjuvant sunitinib treatment (Fig 2C). In a separate study using the identical cell line and implantation protocol, SCID mice bearing orthotopic breast tumors were treated for a shortened period (7 days) with neoadjuvant sunitinib at a higher dose (120 mg/kg). This condensed neoadjuvant treatment protocol yielded similar significant reductions in primary tumor volume and weight (Fig 2D and E, respectively) as compared to controls, with an improved post-surgical survival (Fig 2F). Shorter (7 days) higher-dose (120 mg/kg/day) neoadjuvant sunitinib treatment showed significantly improved survival compared to sunitinib administered in lower doses over a longer period (60 mg/kg over 14 days, respectively) (Fig 2G). Interestingly, similar observations were made in the same model with a vascular disrupting agent, OXi4503. When given neoadjuvantly as one high dose (50 mg/kg) on day 1 of the 7 day treatment period, it provided a significant survival advantage over a lower dose (10 mg/kg) given twice in 14 days. These post-surgical differences contrasted with the significant benefits observed in the pre-surgical setting following neoadjuvant therapy (Supplementary Fig S3A–G). These results confirm that primary tumor response to neoadjuvant antiangiogenic therapy can be highly divergent in predicting post-surgical improvements in survival, and treatment dose, duration, and surgical timing are critical parameters influencing the predictive potential of primary (pre-surgical) neoadjuvant response benefit. Short-term, high-dose sunitinib or VDA therapy may offer improved post-surgical outcomes compared to longer-term, lower-dose treatment.

### Benefits of neoadjuvant therapy before and after surgery depend on the mode of VEGF pathway inhibition

Current clinical trials involving neoadjuvant therapy and VEGF pathway inhibition include small molecule VEGFR TKIs as well as drugs that block VEGF binding its receptor (typically to VEGF receptor 2). Recent preclinical comparisons of protein-based inhibitors of extracellular binding of VEGF ligand to VEGFRs suggest differential benefits in primary versus metastatic disease when compared to VEGFR TKIs. This may depend on model, drug type, and dose used (Paez-Ribes *et al*, 2009; Chung *et al*, 2012; Cooke *et al*, 2012; Sennino *et al*, 2012; Guerin *et al*, 2013), but this has not been examined in the neoadjuvant setting where the effects of pre-surgical

---

**Figure 2. Modulating neoadjuvant sunitinib dose and surgical timing can improve post-surgical survival.**

A–C    SCID mice implanted with LM2-4$^{LUC+}$ human breast cancer cells in the mammary fat pad and treated with three neoadjuvant regimens: vehicle, sunitinib (60 mg/kg/day), or CTX MTD (100 mg, 3 times weekly) for 14 days. (A) Comparison of tumor volume by caliper measurement, and (B) comparison of tumor weight following surgery (36 days post-implantation), with images of excised tumors shown (side panel). (C) Post-surgical survival following neoadjuvant CTX MTD or sunitinib treatment.

D–G    SCID mice implanted with LM2-4$^{LUC+}$ human breast cancer cells in the mammary fat pad and treated with vehicle or sunitinib (120 mg/kg/day) for 7 days. (D) Comparison of tumor volume by caliper measurement, and (E) comparison of tumor weight following surgery (30 days post-implantation), with images of excised tumors shown (side panel). (F) Post-surgical survival following short-term (high-dose) sunitinib treatment compared to control. (G) Post-surgical survival comparison of short-term sunitinib treatment at either high (120 mg/kg/day) or lower (60 mg/kg/day) doses.

Data information: Symbols and bars for box and whiskers plot: median (line), upper/lower quartile (box), min/max (error bars). Survival analysis: hazard ratio (HR), confidence interval (CI), overall survival (OS) based on Kaplan–Meier or Cox regression analysis. *N* = 8–15 mice per group. Neoadj. Tx, neoadjuvant treatment; **P < 0.01, ***P < 0.001 compared to control.

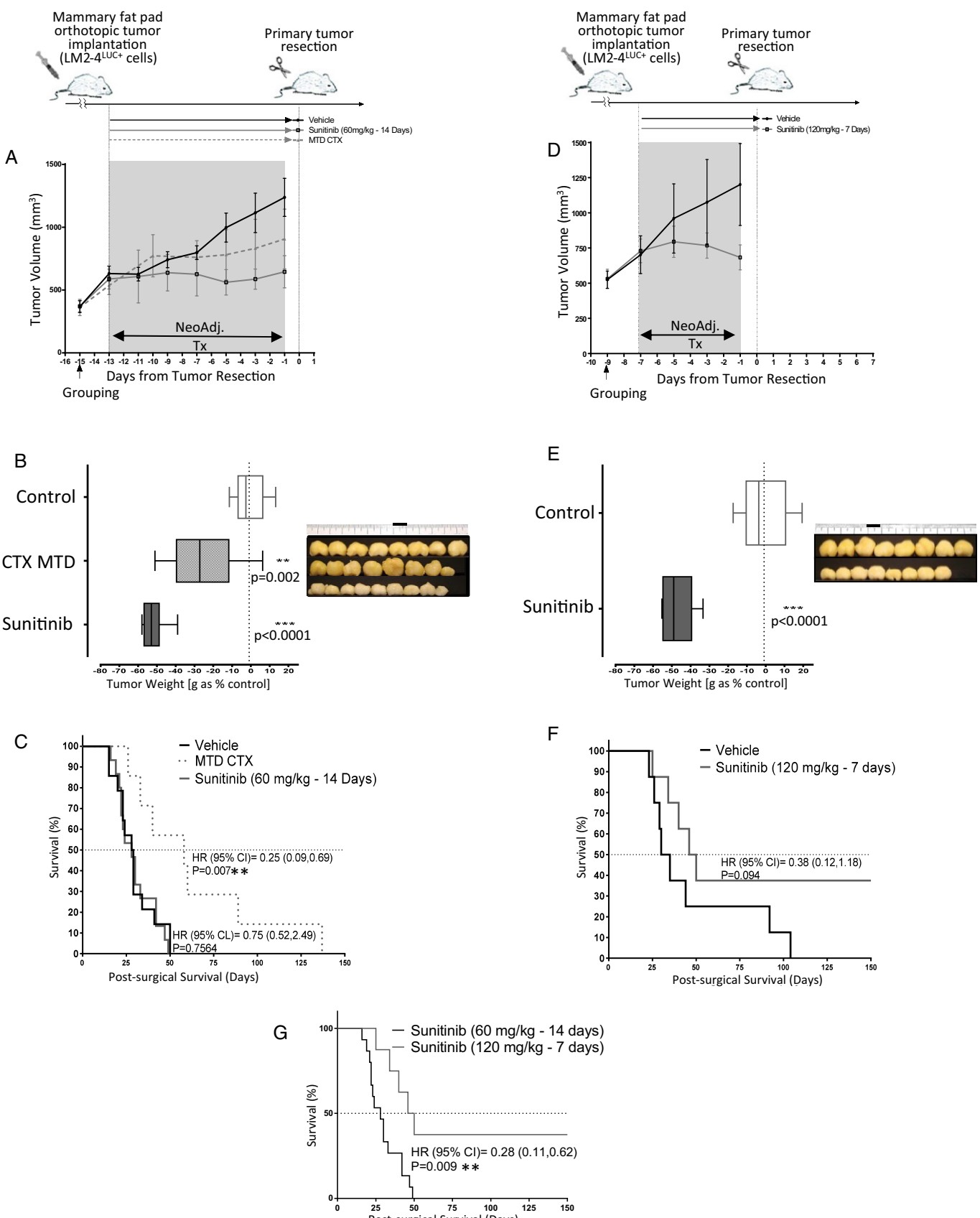

**Figure 2.**

primary tumor response can be compared directly (i.e., in the same mouse) to post-surgical metastasis and survival. We undertook neoadjuvant treatment comparisons of protein-based and TKI-based VEGF/VEGFR inhibition with multiple drugs. This included two VEGF neutralizing antibodies (B20 and G6.31), a VEGFR-2 blocking adnectin (CT322), and two VEGF RTKIs, sunitinib or axitinib. Animals bearing orthotopic LM2-4$^{LUC+}$ tumors were treated for 14 days prior to surgical resection of the primary tumor (see Supplementary Table S1 for treatment schedule and dosing). Significant differences in excised tumor weight were observed in all groups (Fig 3A) compared to vehicle-treated controls. However, only extracellular VEGF/VEGFR-2 inhibitors G6.31, B20, and CT322 showed significant benefits in post-surgical survival compared to control following surgery and treatment cessation, with no improvements observed following sunitinib or axitinib therapy (Fig 3B). Similar comparisons between VEGF RTKIs and antibodies were performed in human melanoma, human kidney, and mouse kidney tumor models. The cells and treatments included LM2-4$^{LUC+}$ (sunitinib, axitinib, and B20), WM113/6-4L (sunitinib and B20), RENCA$^{LUC}$ (sunitinib and axitinib), and SN12-PM6$^{LUC+}$ (sunitinib only). Data from a minimum of two models were analyzed together by standardizing individual mouse data to respective vehicle-treated controls (see Materials and Methods for details). This allowed for graphing to be depicted as having pro- or anti-primary (pre-surgical) tumor benefits and pro- or anti-metastatic (post-surgical) benefits, where overall survival is used as a surrogate for metastasis (see Fig 3C, top left panel for illustration). Grouped analysis of mice from multiple models confirmed that VEGF RTKI and VEGF antibody neoadjuvant treatments yielded pre-surgical anti-primary benefits, with translation to significant post-surgical anti-metastatic benefits only observed in the VEGF antibody groups (Fig 3C and D). Interestingly, grouped analysis of pooled pre- and post-surgical data allowed for evaluation of general treatment–response trends (see Materials and Methods for details). Spearman rank correlation analysis showed that primary tumor benefits (as compared to control) were significantly correlated to overall outcomes in axitinib-, sunitinib-, and B20-treated animals (Fig 3D). This suggests that the magnitude response of the primary tumor at time of resection (as measured by comparing to vehicle-treated group averages) following neoadjuvant treatment may serve as a predictor of overall post-surgical benefits, independent of whether benefit was observed as a group in the pre- or post-surgical setting. Taken together, our results demonstrate that the pre-surgical efficacy of neoadjuvant therapy with an extracellular VEGF inhibitor on the primary tumor is more predictive of post-surgical survival outcomes than VEGFR TKI therapy and that the magnitude of tumor response after neoadjuvant therapy may be an independent surrogate marker of overall post-surgical benefits.

## Tumor-independent host responses to therapy influence experimental metastasis and differ among several treatment types

Since neoadjuvant treatment involves systemic therapy for localized disease, it is possible that non-tumor host responses may influence the extravasive potential of circulating tumor cells prior to surgery. Using a model of experimental metastasis, we administered several anticancer regimens to SCID mice for 7 days prior to i.v. inoculation of human breast (LM2-4$^{LUC+}$) and melanoma (MeWo) cancer cells. Treatments included chemotherapy such as CTX or UFT (a 5-fluorouracil oral prodrug) administered as MTD or low-dose metronomic (LDM) regimens, radiation (XRT), OXi4503, an ALK/c-Met inhibitor crizotinib (PF1066), sunitinib, and several extracellular VEGF pathway inhibitors, including, B20, G6.31, CT322, and DC101—an antibody blocking VEGFR-2 function (see Supplementary Table S1). In the LM2-4$^{LUC+}$ breast cell model, preconditioning mice with XRT, MTD CTX, and sunitinib lead to significant increases in metastasis and a decrease in survival compared to control (Fig 4A), as has been previously shown (Ebos *et al*, 2009; Ebos & Kerbel, 2011). In contrast, a range of outcomes were observed with CT322, G6.31, and B20, OXi4503, LDM UFT, LDM CTX, and LDM CTX/UFT with moderate detriments or improvements in overall survival seen compared to control (Fig 4A). In a similar experiment using MeWo cells, Cox regression analysis was used to stratify pre-treatment effects on eventual survival outcomes compared to control. Therapies listed ranged from favoring control to treatment in the following order: sunitinib > LDM CTX/UFT > crizotinib > CT322 > OXI5403 > DC101 > G6.31 > B20 (Fig 4B). Interestingly, in preconditioning studies in both human breast and melanoma tumor cell models, anti-VEGF antibodies B20 and G6.31 yielded improved overall survival outcome compared to sunitinib monotherapy, but these were not significant. Taken together, various anticancer therapies demonstrate a range of increased or decreased survival compared to control, with extracellular VEGF pathway inhibitors showing more benefit than intracellular TKI-based therapy. Identifying systemic 'host' responses to therapy which facilitate metastasis in an experimental metastasis model could explain potential differential outcomes with therapy in a systemic neoadjuvant treatment setting.

▶

**Figure 3. Pre-surgical effects of neoadjuvant protein-based VEGF pathway inhibition predict for improved post-surgical survival in multiple metastasis models.**

A   Comparison of excised orthotopic LM2-4$^{LUC+}$ breast tumor weights following 14-day neoadjuvant therapy with VEGF RTKIs (sunitinib or axitinib), protein-based neutralizing antibodies to VEGF (G6.31 or B20), or a VEGFR-2 blocking adnectin (CT322).

B   Forest plot summary of post-surgical Cox regression survival analysis following neoadjuvant treatment cessation for groups described in (A).

C   Combined analysis of pre- and post-surgical effects to assess effects on primary tumor and metastatic growth following 14 days of neoadjuvant treatment with sunitinib, axitinib, or B20. Models include LM2-4$^{LUC+}$ (red circle), WM113/6-4L (green square), SN12-PM6$^{LUC+}$ (blue triangle), and RENCA$^{LUC}$ (purple diamond).

D   Corresponding values for primary tumor burden, survival hazard ratio, and Spearman coefficient analysis (see Materials and Methods for details).

Data information: Symbols and bars for box and whiskers plot: median (line), upper/lower quartile (box), min/max (error bars). Survival analysis: hazard ratio (HR), confidence interval (CI), overall survival (OS) based on Kaplan–Meier or Cox regression analysis. $N$ = 6–15 mice per group. Treatment (Tx), vehicle control (Veh). Open symbols were used to indicate data points from animals that were still alive when the experiments were terminated. Crossed lines represent the standard deviation of the vehicle-treated (gray cross) and drug-treated (black cross) from primary tumor burden data (vertical line) and median survival data (horizontal line). *$P$ < 0.05, **$P$ < 0.01, ***$P$ < 0.001 compared to control.

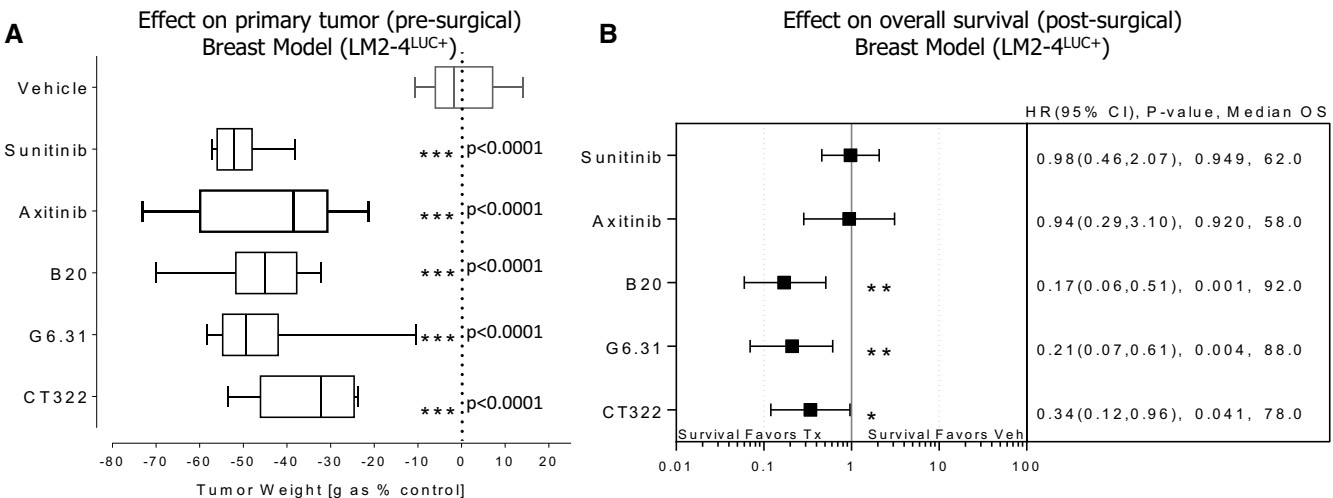

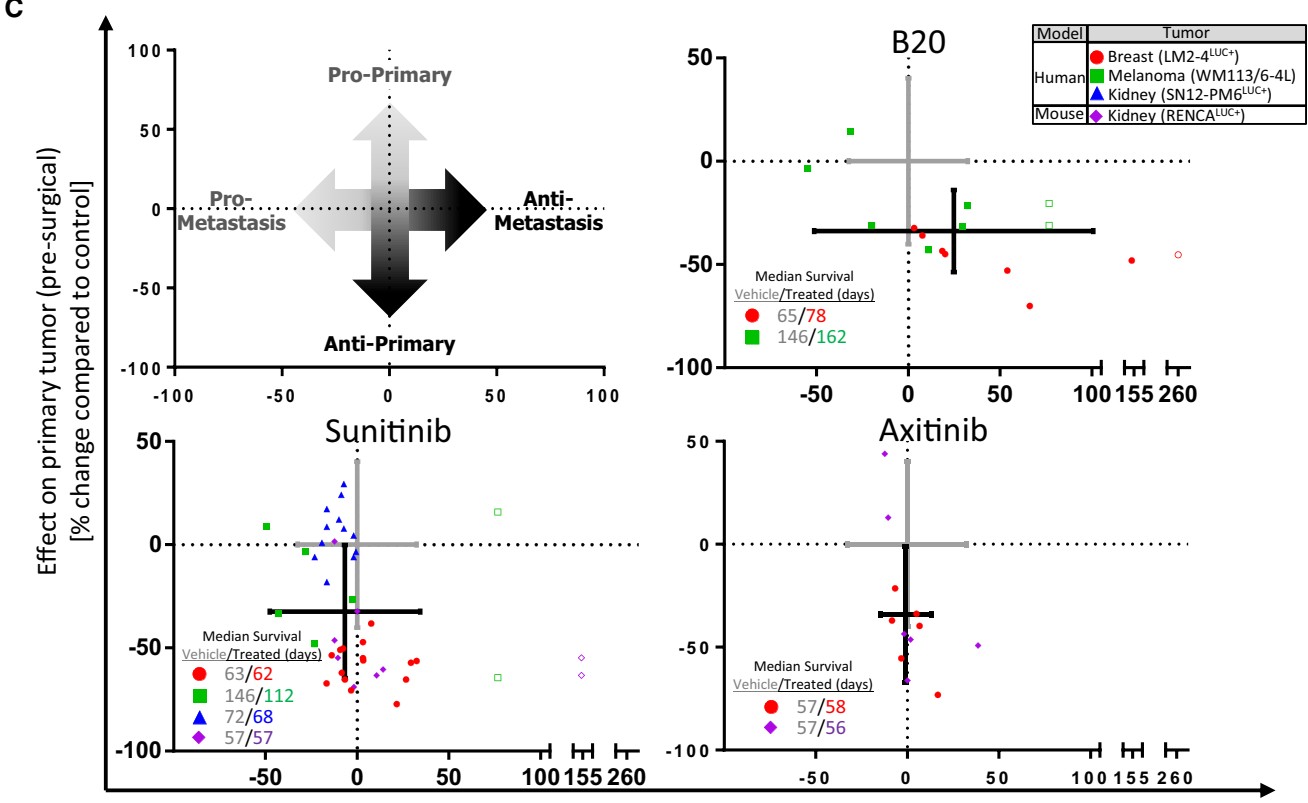

**Figure 3.**

| Drug – Dose – Duration | Pre-Surgical | | Post-Surgical | | Pre- & Post- Surgical | |
| --- | --- | --- | --- | --- | --- | --- |
| | Primary tumor burden (% change) | | Hazard Ratio | | Spearman Coefficient | |
| | | p-value | | p-value | | p-value |
| Sunitinib – 60mg/kg – 14 days | -32.434 | <0.0001 (***) | 1.244 | 0.366 | -0.437 | 0.002 (**) |
| Axitinib – 100mg/kg – 14 days | -34.091 | 0.002 (**) | 1.179 | 0.698 | -0.657 | 0.010 (*) |
| B20 – 5mg/kg – 14 days | -33.778 | <0.0001 (***) | 0.421 | 0.038 (*) | -0.493 | 0.026 (*) |

## Anti-metastatic effects of low-dose chemotherapy may improve neoadjuvant sunitinib treatment

We have previously demonstrated that LDM chemotherapy can significantly prolong the survival of mice with spontaneous metastatic disease, but the same therapy had little to no effect when administered to the same tumor cell line grown as a localized, primary tumor (Munoz *et al*, 2006). These divergent effects in primary and metastatic disease for LDM therapy were observed in the LM2-4$^{LUC+}$ breast model, with a combination of orally administered CTX and UFT either before or after breast tumor removal (Munoz *et al*, 2006). We sought to determine whether neoadjuvant treatment with the LDM CTX/UFT doublet regimen would

recapitulate these findings in a clinically relevant perioperative model. As predicted, primary tumor growth reduction was not observed (Fig 5A) following LDM CTX/UFT therapy, yet following tumor removal and treatment cessation, treated mice had improved overall survival (Fig 5B). In a separate study, we confirm previous observations (Cruz-Munoz *et al*, 2009) that LDM CTX and VBL can have no effect in slowing orthotopic human melanoma tumors (Fig 5C). Interestingly, administration of sunitinib (60 mg/kg) yielded non-significant trends in decreasing overall post-surgical survival but, when combined with LDM CTX/VBL for 14 days, trends toward improved pre-surgical and post-surgical survival were observed (Fig 5D for survival curves; 5E for pre- and post-surgical benefit comparison). These results suggest that the previously

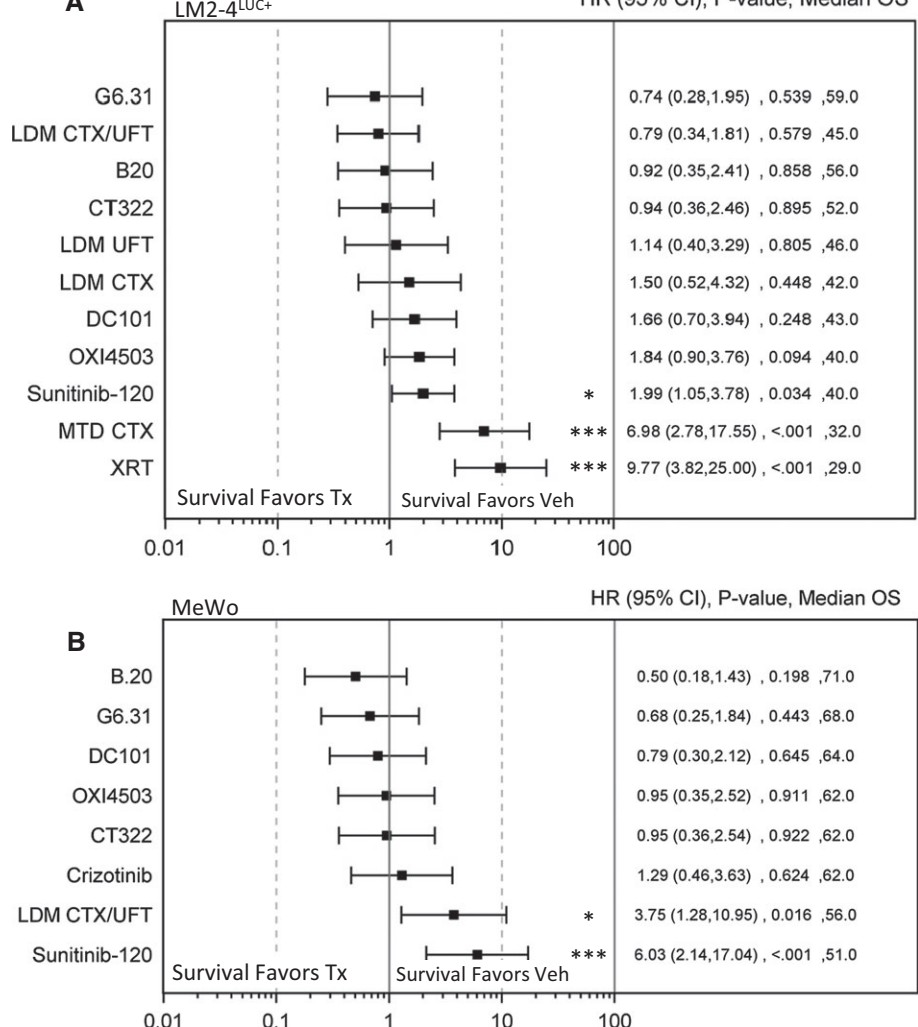

**Figure 4.   Short-term treatment 'preconditioning' in an experimental metastasis model shows range of host-mediated effects on survival.**

A, B   Numerous anticancer treatments, including multiple VEGF pathway inhibitors, were administered during a 7-day period prior to i.v. tumor inoculation to evaluate effects on overall survival. Cox regression survival analysis summary for experiments in SCID mice inoculated with human breast LM2-4$^{LUC+}$ cells (1.5 × 10$^6$ cells) (A) and human melanoma MeWo cells (1 × 10$^6$ cells) (B). Treatments and doses include XRT (5 Gy/1×), MTD CTX (100 mg/kg/3×), LDM CTX (20 mg/kg/DW), LDM UFT (15 mg/kg/D), LDM CTX/UFT, OXI4503 (50 mg/kg/1×), sunitinib-120 (120 mg/kg/D), DC101 (800 μg/3×), CT322 (100 mg/kg/3×), B20 (5 mg/kg/3×), G6.31 (5 mg/kg/3×), crizotinib (50 mg/kg/D). Terms used: three times (3×), one time (1×), drinking water (DW), daily (D), hazard ratio (HR), treatment (Tx), vehicle control (Veh), radiation (XRT), maximum tolerated dose (MTD), low-dose metronomic (LDM), and cyclophosphamide (CTX). For treated groups in (A), N = 7–10 mice, for vehicle groups, N = 29 mice. For all groups in (B), N = 7–8 mice. Survival analysis: hazard ratio (HR), confidence interval (CI), overall survival (OS) based on Kaplan–Meier or Cox regression analysis. See Supplementary Figs S4 and S5 for individual survival curves. *P < 0.05; ***P < 0.001 compared to control.

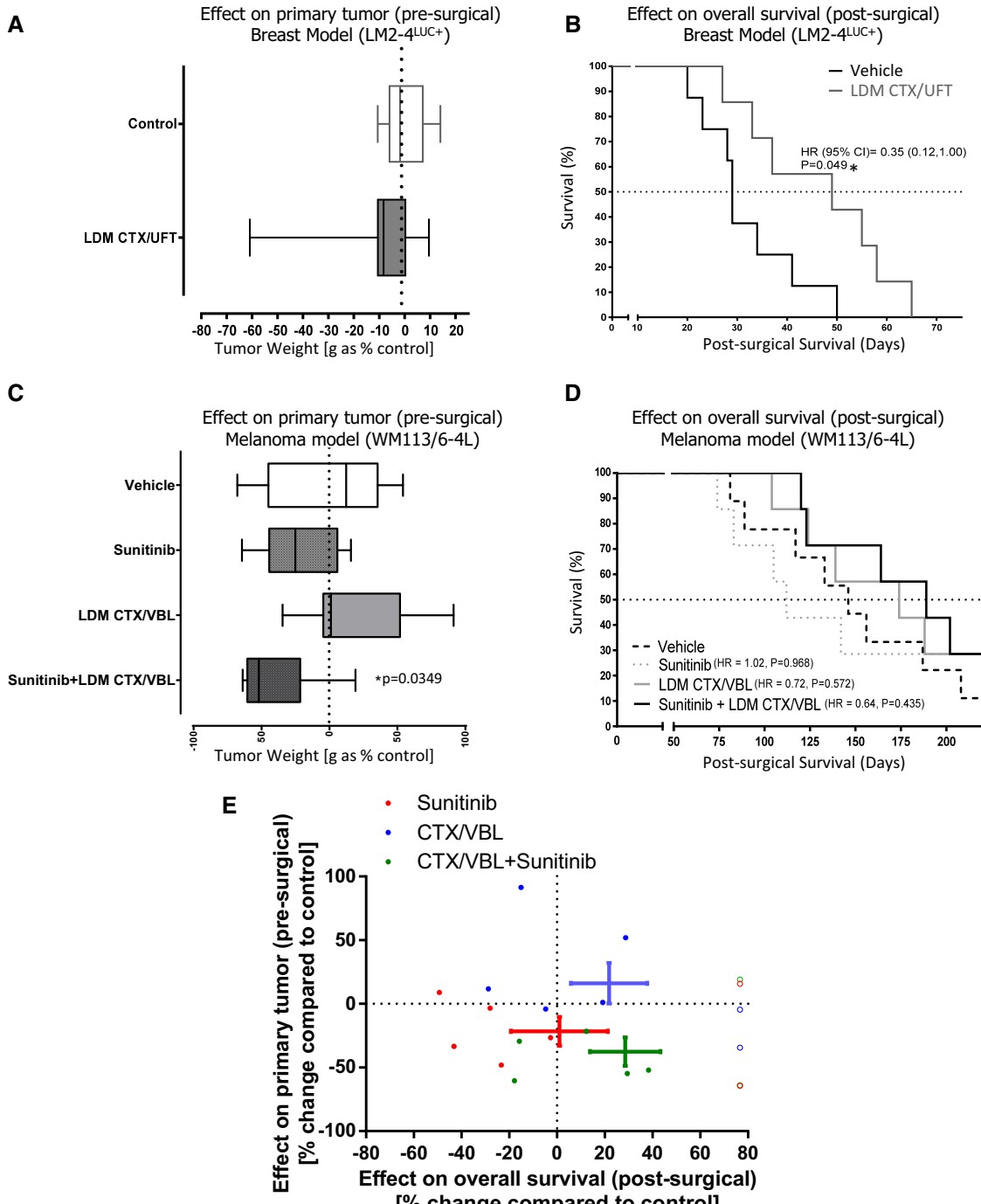

**Figure 5. Rational combination of sunitinib and low-dose chemotherapy can improve neoadjuvant benefits before and after surgery.**

A    Comparison of orthotopic LM2-4[LUC+] breast tumors weights surgically removed following neoadjuvant therapy with LDM CTX/UFT combination for 14 days.

B    Following treatment cessation and surgical resection of primary tumor from (A), comparison of post-surgical survival following LDM CTX/UFT treatment compared to vehicle controls.

C    In a similar comparison, mice bearing WM113/6-4L orthotopic melanoma tumors were given neoadjuvant treatment with neoadjuvant LDM CTX/VBL, sunitinib (60 mg/kg/day), or a combination for 14 days and tumor weights compared following surgical resection.

D    Comparison of post-surgical survival in mice from (C).

E    Combined analysis of pre- and post-surgical effects shown in (C) and (D) to assess effects on primary tumor and metastatic growth following 14 days of neoadjuvant treatment with LDM CTX/VBL, sunitinib (60 mg/kg/day), or a combination of both together.

Data information: Symbols and bars for box and whiskers plot: median (line), upper/lower quartile (box), min/max (error bars). Survival analysis: hazard ratio (HR), confidence interval (CI), overall survival (OS) based on Kaplan–Meier or Cox regression analysis. $N = 7$–9 mice per group. Open symbols were used to indicate data points from animals that were still alive when the experiments were terminated *$P < 0.05$ compared to control.

reported 'anti-metastatic' properties of low-dose metronomic chemotherapy regimens may also include the prevention of micro-metastatic disease following short-term pre-surgical therapy and may reverse the muted or worsened post-surgical effects observed in previously described neoadjuvant treatments with sunitinib.

### A preclinical neoadjuvant efficacy score (NES) to compare perioperative treatment benefits

To better understand the pre- and post-surgical effects of neoadjuvant therapy, we sought to develop a descriptive model to compare (i) pre-surgical effects on the primary tumor, and (ii) post-surgical outcomes. Using excised tumor weights and median survival ratios from experiments listed in Figs 1–3 and 5 (all standardized to vehicle-treated control groups), we established a neoadjuvant efficacy score (NES) to generate a value of overall benefit of therapy as 'anti-primary' (pre-surgical) and 'anti-metastatic' (post-surgical) (see Materials and Methods). Importantly, NES values allow assessment of overall therapeutic impact and identification of neoadjuvant benefit or detriments (Fig 6A). In LM2-4$^{LUC+}$, WM113/6, and RENCA$^{LUC+}$ models, VEGF RTKIs sunitinib or axitinib showed low NES values because anti-primary effects did not translate into anti-metastatic effects, something that was in contrast to protein-based VEGF therapy, which showed improved overall NES values (LM2-4$^{LUC+}$ and WM113/6 models only) (Fig 6B). Furthermore, comparisons of varied neoadjuvant treatment durations and doses can improve NES values. Short-term VDA (OXi4503) (50 mg/kg, one dose) and sunitinib (120 mg/kg/day) treatment over 7 days (NES 1.33 and 0.73) were superior to the same treatments at lower doses over 14 days (NES 0.50 and 0.57, respectively, see Fig 6B upper panel). In the WM113/6 model, NES values for LDM CTX/VBL and sunitinib combined therapy (NES 0.67) show improvement over sunitinib and LDM CTX/VBL treatment alone (NES −0.01 and 0.03, respectively). Intriguingly, and based on the experiments with sunitinib treatment in the SN12-PM6$^{LUC+}$ model described in Fig 1, low NES values demonstrate the lack of overall benefit in this neoadjuvant treatment strategy in both primary and metastatic disease (NES −0.12). Importantly, no consistent trends were observed that suggest neoadjuvant treatment influenced a preferred location of eventual metastasis compared to vehicle-treated controls (see Supplementary Results and Supplementary Fig S6). Taken together, the use of descriptive NES values offers the potential to serve as a predictor of anti-primary and anti-metastatic efficacy, as well as to serve as a tool to compare treatments and predict drug combination strategies to improve overall outcome.

## Discussion

In the present study, we have identified the optimal surgical parameters necessary to examine short-term pre-surgical neoadjuvant treatment effects on primary tumor growth and its influence on (post-surgical) metastatic recurrence after therapy cessation. Multiple therapeutic strategies were used, including protein-based inhibitors of the VEGF ligand or receptor binding, small molecule drugs targeting intracellular VEGFRs (e.g., VEGFR TKIs), vascular disrupting agents, as well as chemotherapy. Herein, we demonstrate that the pre-surgical benefits of neoadjuvant therapy do not

consistently predict for post-surgical disease recurrence and survival, with correlations dependent on treatment dose, surgical timing, treatment duration, and mode of VEGF pathway inhibition. Preclinical neoadjuvant models can be used to uncover (and differentiate) between 'anti-primary' and 'anti-metastatic' treatment effects, and potentially uncover rational treatment combination strategies to improve perioperative outcomes.

We previously demonstrated that short-term treatment with sunitinib prior to intravenous (i.v.) inoculation of breast and melanoma cells could accelerate metastasis and shorten survival, despite cessation of treatment (Ebos et al, 2009). This, along with another similar study (Paez-Ribes et al, 2009), raised the possibility that VEGF pathway inhibition may change the natural history of tumor progression after antiangiogenic therapy and include potential metastasis-promoting effects. However, it remains an open question of how clinically relevant these preclinical findings are (Miles et al, 2010; Blagoev et al, 2013), particularly since there is growing evidence that differential efficacies of anti-VEGF pathway inhibitors extend to not only disease stage (i.e., primary tumor versus micro- and macro-metastatic disease) but also among treatment types (i.e., TKIs versus antibodies). For the latter question, mode of VEGF pathway inhibition may play a key role in explaining different outcomes clinically and preclinically (Ebos et al, 2009; Singh & Ferrara, 2012; Guerin et al, 2013). For instance, it has been recently shown that antibodies neutralizing VEGF do not increase experimental lung metastasis in mice preconditioned to therapy, suggesting these effects may be specific to VEGFR TKIs only and may be dose dependent (Chung et al, 2012; Cooke et al, 2012; Welti et al, 2012). Our results in an experimental model of metastasis where mice were preconditioned by therapy confirm this differential effect and extend these results to compare inhibitors of VEGFR-2 and a VDA, along with direct comparisons to radiation and chemotherapy (including MTD and LDM). We found that VEGFR TKIs (along with chemotherapy and radiation) can lead to decreases in overall survival, but differ from other VEGF inhibition strategies, where no detrimental effects on survival were observed. Critically, despite this difference, protein-based VEGF/R inhibitors (G6.31, B20, CT322, and DC101) still did not result in a significant survival benefit in experimental metastasis models, something not tested in previous publications (Chung et al, 2012). This could explain why sunitinib-induced benefits in primary tumor reduction following neoadjuvant treatment did not translate into post-surgical survival in all instances, and why similar antibody treatment led to improvements in overall survival. One important consideration is that the half-life of biologics such as antibodies is significantly different than for small molecule drugs (2 weeks vs. 12–24 h, respectively). This means target inhibition likely persists long after the neoadjuvant treatment window (something observed with neoadjuvant bevacizumab clinically (Starlinger et al, 2012)) and makes preclinical comparisons with TKIs challenging to assess in the perioperative setting.

For our studies, we chose to evaluate the neoadjuvant setting to determine whether these putative 'pro-metastatic' treatment effects could be observed in a clinically relevant model. In this regard, neoadjuvant therapy could potentially allow for testing of off-target 'host' effects (since it involves systemic treatment) and allow differentiation between primary tumor responses and post-surgical disease recurrence following treatment cessation. Critically, our neoadjuvant therapy model also has the potential to predict drug

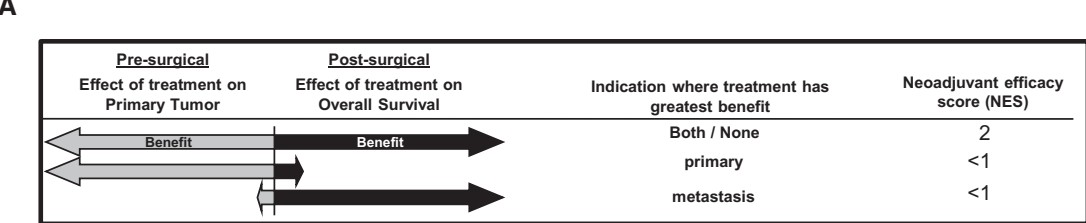

**A**

**B**

Figure 6. Comparison of primary tumor and overall survival benefits to define neoadjuvant efficacy score.

A  Neoadjuvant efficacy score (NES) was defined as the difference between measured benefit of primary tumor response to therapy, and overall survival (OS) benefit after therapy is stopped and tumor resected. All values are compared to control with negative (−) and positive (+) indicating improvement or detriment for primary tumor benefit, respectively, and vice versa for overall survival benefit.

B  NES values for all therapies tested in breast, melanoma, and kidney models shown in Figs 1–3 and 5. See Materials and Methods for NES value determination details.

combinations by uncovering therapies that may yield benefits in primary tumor reduction or metastatic prevention, but not necessarily both. Indeed, therapies with only 'anti-metastatic' properties raise the importance of testing drugs in mouse models where these effects would not be overlooked (such as only testing in localized disease) (Palmieri *et al*, 2007; Steeg *et al*, 2009). In this regard, our

results confirm the potential 'anti-metastatic' effects of LDM doublet combination of UFT/CTX and raise the potential for future use in the pre-surgical setting. It is possible that LDM co-administration may abrogate any negative effects of TKI therapy, if observed. Interestingly, current clinical trials with sunitinib in combination with LDM chemotherapy are now under evaluation in several late-stage

disease types, including in breast (NCT00513695) and in pediatric tumors (Navid *et al*, 2013), as well as in the neoadjuvant setting in RCC (with cyclophosphamide) (Khattak *et al*, 2012).

One key rationale for the use of neoadjuvant chemotherapy in breast cancer patients is that pCR observed in the primary tumor may have value in predicting future (post-surgical) benefit (by delaying recurrence) (Fumagalli *et al*, 2012). It may also have the advantage of guiding clinicians' choice of post-surgical therapy in the case of recurrence if a potent initial response is observed (von Minckwitz *et al*, 2012b). However, there is no preclinical evidence to support this relationship for antiangiogenic therapy nor have the treatment variables such as dose, treatment duration, and OST been tested for their potential to influence post-surgical survival outcomes in models of spontaneous (post-surgical) metastatic disease. In terms of dose, our findings showing that different concentrations of sunitinib and OXi4503 can improve initial differential effects between pre- and post-surgical treatment effects could be of immediate clinical importance. This follows previous preclinical studies demonstrating that higher VEGFR TKI dosing in mice can improve efficacy when compared with the same drug given in lower concentrations, more frequently (Wang *et al*, 2011). Clinically, the use of sunitinib given to patients in doses of 50 mg daily for an intermittent 4 weeks on/2 weeks off schedule showed no improvement when given in a continuous, lower dose (37.5 mg daily, with no breaks) (Motzer *et al*, 2012). Indeed, higher sunitinib dosing is currently under study in mRCC patients with progressive disease if toxicity is tolerated (Bjarnason *et al*, 2013; Pili *et al*, 2013). Our results showing increased dose and shortened surgical window overcoming putative negative (or negligible) post-surgical impact on overall survival could warrant consideration in clinical neoadjuvant trials with VEGFR TKIs, where parameters of tumor dosing and tumor size are still being investigated in terms of assessing overall benefit (Kroon *et al*, 2013). Already evidence from retrospective studies investigating pre-surgical cytoreductive sunitinib treatment in RCC suggest that parameters of treatment stage (Bex *et al*, 2011) and primary tumor reduction (Abel *et al*, 2011) may play a significant role in patient outcomes. In this regard, our results demonstrating that the magnitude of primary tumor response following neoadjuvant therapy correlates with overall survival could support these findings. Furthermore, it is also possible that alterations in standard pre-surgical dosing could alleviate concerns about potential break periods, or gaps in treatment, that typically occur in patients receiving neoadjuvant therapy (e.g., because of toxicity). Related to this, recent retrospective studies in RCC patients receiving pre-surgical VEGFR TKIs showed an increase in proliferative tumor endothelial cells (ECs) in those patients who had a longer treatment break before surgery (Ebos & Pili, 2012; Griffioen *et al*, 2012). But the same study showed that bevacizumab did not yield similar elevations in proliferating ECs. In our studies, we found that elevations in proliferating tumor cell populations in the resected primary tumor following neoadjuvant therapy (as measured by Ki67 levels) may correlate with post-surgical survival benefits. Interestingly, we found that increases in tumoral Ki67 following neoadjuvant B20 and CT322 treatment predicted for decreased survival, whereas the opposite was observed following sunitinib treatment with elevated Ki67 levels predicting for prolonged survival (see Supplementary Results and Supplementary Fig S7). The basis for this difference is currently unknown but could merit further investigations that

examine whether treatment gaps can influence post-surgical survival and/or metastatic disease distribution in the neoadjuvant setting.

Taken together, while traditional cytotoxic treatments (such as chemotherapy) in the neoadjuvant setting have typically resulted in improved survival following surgical intervention, similar benefits with antiangiogenic therapy remain largely untested. Herein, we have identified a novel methodology for evaluating neoadjuvant efficacy using a spontaneous surgical metastasis model and show how it can be used to explain differential efficacies of VEGF pathway inhibitors seen preclinically and clinically. These findings may be immediately relevant to numerous perioperative trials underway in patients.

# Materials and Methods

### Drugs and doses used

Drugs used in this study include SU11248/sunitinib malate, AG013736/axitinib, and an ALK/c-Met inhibitor, crizotinib/PF1066 (Pfizer Inc, New York); UFT, a 5-fluorouracil pro-drug (Taiho, Japan); anti-VEGF antibodies, G6.31 and B20 (Genentech, Roche); targeted adnectin inhibitor of VEGFR-2, CT322 (Adnexus, Waltham, MA); anti-VEGFR-2 antibody, DC101 (ImClone Systems/Eli Lilly, New York); vascular disrupting agent (VDA), OXi4503 (Oxigene, San Francisco, CA); cyclophosphamide (CTX) (Baxter Oncology GmbH, Mississauga, Ontario, Canada); vinblastine sulfate (VBL). All doses, treatment durations, and formulations are summarized in Supplementary Information.

### Mouse tumor models

Animal tumor model studies were performed in strict accordance with the recommendations in the Guide for Care and Use of Laboratory Animals of the National Institutes of Health and according to guidelines of the Canadian Council on Animal Care. Protocols used were approved by the Sunnybrook Health Sciences Centre Animal Care Committee (for studies using LM2-4$^{LUC+}$, SN12-PM6$^{LUC+}$, MeWo, and WM113/6-4L cell lines) and by the institutional Animal Care and Use Committee (IACUC) at Roswell Park Cancer Institute (for studies using LM2-4$^{LUC+}$ and RENCA$^{LUC+}$; Protocol: 1227M).

#### *Experimental metastasis assays*
LM2-4$^{LUC+}$ ($1.5 \times 10^6$ cells) or human MeWo melanoma ($1 \times 10^6$ cells) were injected directly into the tail vein of 6- to 8-week-old female CB-17 SCID mice (Charles River, Canada) as previously described (Ebos *et al*, 2009).

#### *Human xenograft and mouse syngenic orthotopic tumor implantation and primary tumor resection*
LM2-4$^{LUC+}$ cells ($2 \times 10^6$ cells in 50 μl), WM113/6-4L ($1 \times 10^6$ cells in 150 μl), SN12-PM6$^{LUC+}$ ($2 \times 10^6$ cells in 5 μl), and RENCA$^{LUC+}$ ($4 \times 10^4$ cells in 5 μl) were orthotopically implanted, respectively, into the right inguinal mammary fat pads (right flank), dermis (right flank), or kidney (subcapsular space) of 6- to 8-week-old female CB-17 SCID or BALB/c mice as previously described (Munoz *et al*, 2006; Cruz-Munoz *et al*, 2008; Tracz *et al*, 2014). Primary melanoma and breast tumor size was assessed regularly with vernier

calipers using the formula width$^2$ × length × 0.5. Breast and kidney tumor models utilized bi-weekly bioluminescent monitoring, which has previously been demonstrated to parallel overall tumor burden (Ebos *et al*, 2009). Prior to neoadjuvant therapy, mice were grouped according to tumor burden (melanoma and breast according to tumor size, and kidney according to bioluminescence), ensuring that the mean average was not statistically different. Representative examples of neoadjuvant pre-treatment sorting to standardize grouping are shown for SN12-PM6$^{LUC+}$ (Supplementary Fig S8A), RENCA$^{LUC+}$ (Supplementary Fig S8B), WM113/6-4L (Supplementary Fig S8C), and LM2-4$^{LUC+}$ cells (Supplementary Fig S8D). Optimal surgical times (OST) were determined by assessing metastatic potential (MP) following surgery at various time points to determine probability of spontaneous metastasis not derived from local invasion (see Supplementary Fig S1E–G). OST determination was made using vehicle-treated controls from various experiments. No difference in metastatic disease progression patterns or survival has been noted between vehicles or between vehicle and untreated animals. Approximate tumor weights to determine OST were based on previous studies with LM2-4$^{LUC+}$ and WM113/6-4L tumor sizes of ~400 mm$^3$ and ~200 mm$^3$, respectively as previously described (Cruz-Munoz *et al*, 2008; Ebos *et al*, 2009). For kidney tumor model, 1–2 days after cessation of neoadjuvant therapy, kidney nephrectomy was performed. Excised kidneys were examined for encapsulated tumor. If any tumor invasion into the peritoneal space was noted, the mouse was removed from the study. For kidney models, in any rare instance where no tumor was ever present at any time before and after surgery and treatments (determined by BLI or visible macroscopically), mice were excluded from the study so as not to give potential false positive or negative bias to results.

### Defining parameters for establishing a neoadjuvant treatment period

#### Residual cancer burden (RCB)
During surgical resection of primary tumor, any residual or localized tumor invasion was noted and RCB was broadly compared to clinical pathological complete response (pCR). Acceptable RCB was defined as not having visual residual tumor or obvious local invasion. The presence of bioluminescent-positive tumor following surgery (shown in Supplementary Fig S1F) indicated micrometastatic disease, but was considered acceptable since this detection sensitivity exceeds clinical comparisons. Unacceptable RCB was defined as any localized invasion or unresectable primary tumor mass. In melanoma and breast models, skin or peritoneal wall was removed if invasion was noted, with clear margins taken of normal tissue. A mouse was excluded from the study if clear margins were not attainable. In the kidney model, any subcapsular tumors growing outside of the kidney were noted as 'invasive' and ensured to not have invaded surrounding tissue prior to nephrectomy.

#### Justification of timing between treatment cessation and surgery
All treatments in the experimental metastasis model or neoadjuvant perioperative model included treatments that were halted 24 h before surgery. We previously demonstrated therapy cessation in this timeframe (Ebos *et al*, 2009), and pre-surgical sunitinib treatment in RCC patients has included 24 h in certain trials, though this varies among ongoing clinical trials (Schrader *et al*, 2012).

### Neoadjuvant Efficacy Score (NES)

Independent parameters of neoadjuvant efficacy in our studies included primary tumor response (PTR), which compared resected primary tumor weight (RPTW) following treatment to control, and overall survival response (OSR), which compared the median survival after treatment to control. These quantitative measurements were combined as a descriptive analysis of overall treatment benefit as a neoadjuvant efficacy score (NES), with the aim to compare PTR and OSR among treatments. The following equation was used to establish the NES, with indication (i.e., PTR or OSR or none) with the greatest benefit noted.

$$\text{PTR (\%)} = \frac{\text{Treated RPTW (AVG)} - \text{Vehicle RPTW (AVG)}}{\text{Vehicle RPTW (AVG)}} \times 100$$

$$\text{OSR (\%)} = \frac{\text{Treated OS (median)} - \text{Vehicle OS (median)}}{\text{Vehicle OS (median)}} \times 100$$

$$\text{NES} = \frac{\text{OSR} - \text{PTR}}{100}$$

### Statistical analysis

Results were subjected to statistical analysis using the GraphPad Prism software package v.4.0 (GraphPad Software Inc., San Diego, CA), IBM SPSS Statistics v22.0 (IBM Corp., Armonk, NY), and SAS v9.3 (Cary, NC). All growth curves shown are represented as mean ± standard deviation (SD). Overall survival was summarized using the Kaplan–Meier method with the association between treatment group and survival evaluated using the two-sided log-rank test. Cox regression models were used to obtain hazard ratio estimates, with corresponding 95% confidence intervals, for comparing treatment groups to control. Correlation plots for combined pre- and post-surgical summary analysis (Figs 3C and 5E) were conducted as follows. Pre-surgical primary tumor effects: Resected tumor or tumor-bearing kidney weights normalized to control animals following one-sample one-tailed *t*-test comparison. Post-surgical survival effects: HR based on Cox regression analysis; median survival based on Kaplan–Meier analysis. Pre-/post-surgical correlation: Spearman rank correlation used one-tail test for significance (linear regression avoided due to censored data). Student's *t*-tests were one-tailed and unpaired. A minimum significance level of 0.05 was used for all analysis.

**Supplementary information** for this article is available online: http://embomolmed.embopress.org

### Acknowledgements
We would like to thank Dr. Alejandro Godoy and Dr. Gary Smith for valuable technical assistance with staining protocols; Dr. Ping Xu for OXi4503 dosing advice; Dr. Roberto Pili for the RENCA$^{LUC+}$ cells; Petia Stefanova for construction of the TMAs; and several companies for drug. These include Pfizer for sunitinib and axitinib (Dr. J. Christensen); ImClone Systems for DC101 (Dr. Bronek Pytowski); Genentech for B.20 and G6.31 (Dina Washington); Adnexus for CT322 (Eric Furfine); Taiho Pharmaceuticals for UFT (Teiji Takechi). This study has been carried out with grant support from the following: Canadian Breast Cancer Foundation (to R.S.K); the Canadian Institutes of Health and

## The paper explained

### Problem

A hypothesis guiding the clinical development of new cancer therapy is that benefits observed in patients with late-stage metastatic disease may also have benefits in the treatment of earlier cancer stages. With the approval of antiangiogenic drugs that target the vascular endothelial growth factor (VEGF) pathway in the metastatic setting, several trials have been initiated in the pre-surgical setting, which aim to treat a localized primary tumor before it is removed (termed 'neoadjuvant' therapy). Yet there are few, if any, preclinical studies that aim to recapitulate neoadjuvant treatment, and none have involved rigorous comparison of multiple VEGF pathway inhibition strategies. The reason for this is, at least in part, because few preclinical tumor models effectively mimic spontaneous metastatic growth following surgical resection of an orthotopic tumor. Additional challenges include the need to define the optimal time to start/stop neoadjuvant therapy and perform surgery, as well as allow comparisons of different doses and drug types—all of which may influence pre- and post-surgical disease progression.

### Results

In this study, we used four mouse models of spontaneous metastatic disease—each established by first implanting cells orthotopically and then surgically removing the primary tumor. We then identified the optimal surgical parameters necessary to compare effects of short-term pre-surgical neoadjuvant treatment on post-surgical metastatic recurrence after therapy cessation. Our results show that primary tumor reductions following neoadjuvant VEGF RTKIs do not consistently correlate with post-surgical metastatic recurrence rates, which can be offset by reduced survival in some instances. Importantly, we found that negative post-surgical effects could be reversed with increased dose, shorter treatment duration, and earlier surgical times. Interestingly, protein-based VEGF pathway inhibitors (such as antibodies) provide an example of how drug efficacies can differ within drug classes, with survival benefits typically improved.

### Impact

We describe for the first time a novel preclinical methodology to examine the effects of neoadjuvant therapy in a clinically relevant setting. These models can be used as a tool to differentiate 'anti-primary' and 'anti-metastatic' effects. Furthermore, we present a novel scoring system that could serve to predict drug combinations that maximize neoadjuvant VEGF RTKI treatment benefits. This includes altering dose and treatment duration to improve post-surgical outcomes as well as combination with low-dose metronomic chemotherapy regimens shown to have anti-metastatic activity.

Science (to R.S.K); the Ontario Institute for Cancer Research (to R.S.K); and the Roswell Park Alliance Foundation (to J.M.L.E).

## Author contributions

Study conception and design (JMLE, CRL, MM, RSK). Acquisition of data (JMLE, CRL, MM, AT, WRC, CJ). Analysis and interpretation of data (JMLE, CRL, MM, AT, JMH, KA, PB, WRC, CJ). Manuscript writing (JMLE, CRL, MM, RSK).

## Conflict of interest

The authors declare that they have no conflict of interest.

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
