## [Review Process File · EMBO Molecular Medicine]

Neoadjuvant antiangiogenic therapy reveals contrasts in primary and metastatic efficacy

John M.L. Ebos, Michalis Mastro, Christina R. Lee, Amanda Tracz, John M. Hudson, Kristopher Attwood, William R. Cruz, Christopher Jedszko, Peter Burns, and Robert S. Kerbel

Corresponding author: John Ebos, Roswell Park Cancer Institute

Review timeline:

Submission date:	19 February 2014
Editorial Decision:	25 March 2014
Revision received:	02 September 2014
Editorial Decision:	11 September 2014
Revision received:	23 September 2014
Additional Editorial Correspondence:	24 September 2014
Additional Author Correspondence:	25 September 2014
Accepted:	25 September 2014

Transaction Report:

Editor: Roberto Buccione

1st Editorial Decision

25 March 2014

Thank you for the submission of your manuscript to EMBO Molecular Medicine. We have now heard back from the three Reviewers whom we asked to evaluate your manuscript.

We apologise that it has taken more time than we would have liked to return a decision, but unfortunately one Reviewer delivered his/her evaluation with some delay.

You will see that while all three Reviewers are supportive of your work, Reviewers 1 and 2 express a number of concerns that prevent us from considering publication at this time. I will not dwell into much detail, as the evaluations are quite detailed and self-explanatory.

Reviewer 1 raises two main issues. The first is a general consideration that the strength of the conclusions is somewhat diminished by the fact that only immunodeficient mouse models were used; the second is the lack of statistical significance for many of the conclusions and experimental settings, which poses significant problems in the correct interpretation of the results. Reviewer 1 also challenges the translational potential of the findings. S/he also raises a number of additional issues that require your action.

Reviewer 2, similarly to Reviewer 1, notes the lack of statistical significance in a number of instances. S/he also questions clinical relevance based on the fact that the models used are aggressive ones and thus would not justify, in the clinic, the approach used here. Reviewer 2, as does Reviewer 1, also raises several additional issues that require intervention.

My impression is that Reviewers 1 and 2 are globally quite positive and clearly underline that your work addresses an urgent and very important need. They do, however, raise sensible, if not critical points. I would therefore ask you to make a special effort to address the above concerns scrupulously and with additional experimentation where necessary. I would also ask you to address the model suitability issue as far as realistically possible; of course, further experimentation on additional models as suggested would be ideal. Indeed, if you already have such data, this should be included. On the other hand I understand that this would require extensive and lengthy experimentation and thus, depending on your response to the Reviewers and their evaluation, I would be prepared to forego this requirement provided the other issues are thoroughly addressed.

I also feel that, given the inherent complexity of the experimentation and data generated, an effort should be made to shorten the manuscript and to better focus the message.

While publication of the paper cannot be considered at this stage, we would be pleased to consider a suitably revised submission, with the understanding that the Reviewers' concerns must be fully addressed as mentioned above with additional experimental data where appropriate and that acceptance of the manuscript will entail a second round of review.

Please note that it is EMBO Molecular Medicine policy to allow a single round of revision only and that, therefore, acceptance or rejection of the manuscript will depend on the completeness of your responses included in the next, final version of the manuscript.

As you know, EMBO Molecular Medicine has a "scooping protection" policy, whereby similar findings that are published by others during review or revision are not a criterion for rejection. However, I do ask you to get in touch with us after three months if you have not completed your revision, to update us on the status. Please also contact us as soon as possible if similar work is published elsewhere.

I look forward to seeing a revised form of your manuscript as soon as possible.

***** Reviewer's comments *****

Referee #1 (Comments on Novelty/Model System):

The observations have only been demonstrated in SCID mouse models. If similar effects could be demonstrated in immune competent syngenic tumor models of metastatic disease this would further strengthen the conclusions.

Referee #1 (Remarks):

Summary:

In the manuscript EMM-2014-03989 "Neoadjuvant antiangiogenic therapy reveals contrasts in primary and metastatic efficacy" Ebos and co-workers investigate the potential of modeling the neoadjuvant application of different types and combinations of antiangiogenic and chemotherapy in three previously established postsurgical orthotopic mouse models of advanced metastatic disease. The authors develop the necessary experimental conditions for effective neoadjuvant treatment and surgical resection to ensure metastatic potential as a prerequisite for their subsequent assessment of response on primary tumor growth and postoperative metastatic recurrence. They conclude, that preclinical neoadjuvant tumor models can be used to differentiate between anti-primary and anti-metastatic treatment effects that could lead to rational combination strategies to improve

perioperative outcomes in patients.

General judgment:

Overall, the manuscript is addressing the very interesting and clinically important problem of frequently discrepant therapeutic outcomes observed between preclinical and clinical studies. The frequent failure to predict successful therapeutic strategies for clinical trials on the basis of preclinical efficacy is partly due to the lack of adequate models of advanced metastatic disease given that traditionally most preclinical tumor models only assess primary tumor growth response. The problem has prominently been illustrated by the disappointing results of sunitinib in clinical trials. The authors add to this field by using their unique platform of recently established postsurgical tumor models with spontaneous metastatic disease, to explore the effects of different VEGF pathway inhibitors with and without chemotherapy in the neoadjuvant setting. They show that counter-intuitively the presurgical benefit of neoadjuvant therapy does not consistently predict for postsurgical disease recurrence and survival, which is influenced by a plethora of variables, which this study begins to shed light on. While still relatively preliminary, these studies could potentially offer new insights into the biological differences between primary and metastatic disease and help explain their differential response to therapy.

While the concept of the manuscript is absolutely worthy, the data in the manuscript clearly illustrate the huge obstacles involved in developing appropriate tools to predict outcomes for patients receiving neoadjuvant antiangiogenic therapy. The authors approach is based on a long track record of studying the effects of metronomic chemotherapy and antiangiogenic therapy in various tumor models. A weakness of the manuscript is that the observations have only been demonstrated in SCID mouse models. If similar effects could be demonstrated in immune competent syngenic tumor models of metastatic disease this would further strengthen the conclusions. Also, the translational potential for predicting drug combinations that have an increased likelihood of success in clinical trials is not immediately obvious. If anything, the results clearly show that even for a given cancer model the outcome can be different for primary and metastatic tumors, even when the same treatment is used in different concentrations or application schedules. Whether these data translate to other tumor models or into the clinical setting is uncertain. Regardless of this, there is clear value in this approach in terms of developing and testing a preclinical platform to explore drug regimes in the neoadjuvant setting. These findings should therefore be viewed as a hypothesis generating starting point for further investigations that compare specific drug classes or dosing schedules in order to advance our understanding of the underlying cancer biology and the broader principles of metastatic progression that may eventually be leading to future clinical applications.

Specific Points- Major:

1. Fig. 1E and F: The authors state on Page 7 that there is a reduction in WM113/6-4L tumor volume and tumor weight after 14 d compared to vehicle Fig. 1E/G. There is no statistics to support this and the data does not seem to show any relevant differences (which would fit with the data presented for the RCC model in Fig. 1A/B). Could the authors clarify this?
2. Since Fig. 1G does not reach significance. The general conclusion concerning both kidney and melanoma models is somewhat misleading. In any case would the final conclusion that primary tumor response following neoadjuvant sunitinib therapy failed to predict post-surgical survival not be better supported by the breast cancer model, since Fig. 2A-C show a reduction in primary tumor growth in response to sunitinib treatment that again fails to translate into a postsurgical survival benefit similar to the RCC?
3. Fig. 2F: the conclusion that short term high dose sunitinib may offer improved postsurgical outcomes compared to longer-term, lower dose treatment is intriguing but does not appear to be statistically supported by the data. This is unfortunate, as this could be easily translated into clinical trials.

4. Concerning the metastatic breast CA model (LM2-4 Luc+): in a recent Cancer Research publication by Guerin et al., 2013 the Kerbel group commented on the use of this tumor cell line: "We did not use a luciferase-tagged clone of LM2.4 (19) because we have found that these cells have a reduced ability for spontaneous metastasis (unpublished observations), the basis of which is currently unknown." Could the authors comment on this?

5. Fig. 3 and S3E: Again the statement that neoadjuvant OXI4503 efficacy was 'significantly' improved by short term, high dose treatment is not statistically supported by the data in Fig. S3E.

6. Fig. 5D: The authors acknowledge, that the data in Fig. 5D is not statistically significant. The statements made in the heading of this paragraph and the title of the figure are not statistically supported by the data shown.

7. The authors suggest that the Neoadjuvant Efficacy Score (NES) calculated in Fig. 6B might serve as a predictor of anti-primary or anti-metastatic efficacy and that it could serve as a tool to compare treatments and predict drug combination strategies to improve overall outcome. However, the predictive value of the NES is not obvious, given that the same treatments can have different effects in different tumor models, or depending on dose. Please clarify.

8. The data summarized in Fig. 6B is clearly useful for the purpose of comparing the therapeutic effects and identifying potentially synergistic drug combinations. In particular, the concept would be strengthened if rational combinations for pre- and postsurgical therapy could be predicted on this basis. For example if differential efficacy of high and low dose OXI4503 or Sunitinib could be recapitulated in one of the other tumor models or if Sunitinib low (60mg), which is effective in reducing primary tumor growth could be combined successfully with postsurgically active drugs such as OXI4503 or LDM CTX/UFT.

9. A general weakness of the manuscript is that the observations have only been demonstrated in SCID mouse models. If similar effects could be demonstrated in immune competent syngenic tumor models of metastatic disease this would further strengthen the conclusions.

10. The results in Fig. 7 should be discussed some more.

11. In Fig. S1 A and B it is not clear to me which tumors belong to which primary surgery time point and why the minimum OST threshold was determined at different time points for the different tumor entities. What is the value of the linear regression lines?

12. While the manuscript is generally well written, it could benefit from a more focused comparison of equivalent results between tumor models and a more critical assessment of the data obtained in different tumor models. The figures would benefit if direct comparisons between equivalent data would be made easier, for example by naming the tumor models at the top of each figure and arranging the corresponding data below for each model.

Specific points - minor:

1. Page 10: improved postsurgical survival after short term high dose OXI4503 is documented in Fig. S3E (not D).
2. Fig. 6B: OXI4503 should be 50mg/kg
3. Page 14: "Similar correlations with CD32 were not observed following in SN12-PM6 Luc+..."

Referee #2 (Comments on Novelty/Model System):

See detailed comments to author

Referee #2 (Remarks):

In this manuscript, Ebos, Kerbel and colleagues employ three mouse models of orthotopic and metastatic human cancer (breast, kidney and melanoma) to comprehensively study the effects of neoadjuvant therapy on (i) primary tumor growth; (ii) post-surgery survival and metastatic burden. They employ a large number of therapeutics, primarily antiangiogenic drugs such as a broad-spectrum TKI (sunitinib), a vascular disrupting agent (VDA), and VEGF/R2 blocking antibodies/adnectin. Based on their findings, the authors conclude that post-surgical survival does not consistently correlate with primary tumor responses in the neoadjuvant setting.

The study is of significant interest as it represents an attempt to comprehensively study the effects of distinct types of neoadjuvant treatments - mostly antiangiogenic - on post-surgery survival in mouse models of metastatic cancer. While this study certainly merits publication, I feel that the clinical relevance of some of the models/regimes is questionable and should be clarified. Furthermore, some of the authors' conclusions are at times at odds with the data presented. The authors may consider the following comments and criticisms to clarify some important aspects of their study.

General criticisms

1. Clinical relevance. The authors used 3 aggressive tumor models, which readily metastasize to multiple organs. The choice to perform neoadjuvant therapy followed by a post-surgery observation time is not completely justified. Because the tumors already metastasize during the treatment, it would be interesting to see whether post-surgery adjuvant therapy (same or another drug) would reverse the prometastatic effects seen with certain drugs, or generally improve survival. In a clinical setting, these primary but already metastatic tumors would be treated with a short neoadjuvant treatment (if any) followed by (cycles of) adjuvant therapy.
2. CTX is not the preferred cytotoxic drug for neoadjuvant therapy in breast cancer. A combination of an anthracycline and a taxane would have been most relevant. Although I understand that this study is primarily focused on antiangiogenic drugs, the authors should comment on this point or test the most relevant drugs in a side-by-side comparison.
3. The clinical relevance of neoadjuvant therapy for metastatic melanoma is unclear. The authors should elaborate on this.
4. Some of the results are commented based on data that failed statistical analysis. While I am convinced that "trends" toward differences are worth reporting, the strength of the conclusions is certainly undermined in such cases.
5. It seems that the authors overemphasized in the Abstract and elsewhere in the ms the discrepancies between pathological responses in the primary tumors and mouse survival. Overall, the data seem to show the opposite, i.e., a general correlation between the two (see Figure 6). The authors should revise their statements to more rigorously reflect this fact.

Specific criticisms by Figure

Figure 1. At the end of the first section of the Results, the authors state that "primary tumor response following neoadjuvant therapy DID NOT predict for benefit in post-surgical survival...". However, in both kidney and melanoma models sunitinib (60mg/Kg) achieved neither complete nor partial pathological responses as primary endpoint. So, it is somewhat expected that post-surgical survival be not increased by neoadjuvant treatment in these models. If any, post-surgical survival is slightly decreased in the kidney cancer but not melanoma model (KM, p value is 0.97). A more appropriate conclusion for this first set of experiments should therefore be "LACK OF primary tumor response following neoadjuvant therapy DID predict for LACK OF benefit in post-surgical survival...". Note that panels G and H are incorrectly called in the main text (there is no panel I).

Figure 2. These experiments are, in principle, interesting, particularly the comparison between the 14-day long neoadjuvant sunitinib (60mg/Kg) and the "condensed" 7-day long neoadjuvant sunitinib (120mg/Kg). However, it is somewhat unfortunate that, in the condensed setting, tumors were surgically removed one week earlier than in the other experiment. Is there a specific reason for selecting such schedule rather than a neoadjuvant therapy -> drug withdrawal -> surgery schedule that would overlap with the 14-day schedule? The interpretation of the results may therefore be confounded by this factor, and a direct comparison between the two schedules seems not possible. Also, the improved post-surgery survival in the condensed setting is questionable (panel F). It would be informative to directly compare the low and high dose sunitinib treatment in a single experiment and reassess primary endpoints as well as survival benefits.

Figure 3. The breast cancer data look compelling. I would add the high VDA dose in Figure 3A-B and move the melanoma data (C,D) to the supplemental data.

Figure 4. In their interpretation of the results from this set of experiments (7-day pretreatment schedule followed by iv injection of breast or melanoma cancer cells), the authors should more rigorously consider the significance of the statistical analysis of the data. Indeed, no drug provided survival benefit as monotherapy, contrary to the message conveyed by the authors. Interestingly, aggressive treatments like high dose sunitinib, MTD CTX and RT worsened survival. I believe that this is the main message of this set of data. Furthermore, the clinical significance of the 7-day preconditioning schedule is unclear and should be discussed more thoroughly. Does this mimic a post-surgery adjuvant setting?

Figure 5. These data essentially reproduce published work from these authors that low-dose, metronomic cytotoxic therapy (LDM, here CTX/5FU) improves post-surgery mouse survival but does not induce pathological responses in the primary tumor in a breast cancer model. Panels A-B show the breast cancer model, panels C-D the melanoma model. Only in the latter model was LDM combined with low-dose sunitinib to explore combinatorial effects on the primary tumor and post-surgical survival. However, the data do not show statistically significant differences, so the message of this combination experiment is unclear. It would be very interesting to see the effects of sunitinib + LDM in the breast cancer model, where sunitinib monotherapy is effective on the primary tumor but does not improve survival (Figure 2B-C).

Figure 6. This Figure provides an interesting and comprehensive summary of the results. It is clear from this beautiful representation of the results that in most instances the data suggest concordant responses in primary versus metastatic tumors (i.e., effects on primary tumors and mouse survival). LDM and high-dose sunitinib represent notable exceptions, but apparently only in the melanoma model. Based on these findings, the authors should revise their claim that the benefits observed in primary tumors often do not correlate with overall survival. In general, the data seem to show the opposite.

Figure 7. This Figure is a bit off-topic and may be move to the supplemental data.

Referee #3 (Remarks):

This is an excellent paper that addresses important questions related to neoadjuvant antiangiogenic therapy. By using various models of spontaneous metastasis, the authors have evaluated pre-surgical therapy with regard to primary tumor response and metastasis. They show that primary tumor response and overall survival sometimes do not correlate, and that significant differences exist between different classes of VEGF inhibitors. The results obtained using different therapeutical regimens are complex, hence the authors established a "neoadjuvant efficacy score" which may be helpful to evaluate the efficacy of neoadjuvant therapy. Overall, the paper is an important contribution that may eventually help to improve the design clinical trials.

The preclinical models used in this study are well suited to address the efficacy of neoadjuvant therapy, the results are convincing and well presented. However, the following point needs to be addressed:

- on page 7, the numbering of Figure 1 is partially confused: Figure 1H shows the evaluation of the metastatic sites, and Figure 1I does not exist.

General Comments and Major Changes to Manuscript:

Note: For clarity, we have given reviewer comments numbers (e.g. R1-1 = reviewer 1, comment 1), which we use as a reference in various comment sections below.

1) Addition of an immuno-competent mouse model of metastasis: Reviewers raised concerns regarding the use of only human tumor cell lines in immunodeficient mice (see reviewer comment R1-9). We have now added a mouse syngeneic metastasis model to increase the clinical relevance of our neoadjuvant antiangiogenic therapy evaluations. Using the mouse kidney tumor (RENCA^{LUC+}) spontaneous metastasis model (recently described in (Tracz et al, 2014)), we have now included additional experiments with neoadjuvant Sunitinib treatment. This is included as Figure 1E-1I – replacing the human metastatic melanoma model, which has been moved to Supplemental Figure S2. The RENCA^{LUC+} tumor model is now also included in the revised Figure 3 (which adds Axitinib treatment) and in Figure 6 (see details below). Altogether, this revised manuscript now includes four human and one mouse cell line (LM2-4 breast, SN12-PM6 renal, WM113/L melanoma, MeWo melanoma and RENCA^{LUC+} renal, respectively), with all (except MeWo) used to evaluate spontaneous metastasis following surgical orthotopic implantation and resection of the primary tumor (or nephrectomy as in the SN12-PM6 and RENCA^{LUC+} models). This manuscript now represents the most comprehensive preclinical examinations into the efficacy of neoadjuvant antiangiogenic therapy in a clinically relevant setting to date.

2) Addition of another VEGF RTKI (Axitinib): Our original manuscript included comparisons between one VEGF RTKI (sunitinib) and three extracellular protein-based inhibitors of the VEGF pathway in the neoadjuvant setting (G6.31, B20, CT322). We have now included data using Axitinib, another FDA-approved VEGF RTKI, in two surgical metastasis models testing neoadjuvant treatment (LM2-4 and RENCA). All results with axitinib confirm the overall pre- and post-surgical efficacy trends seen with sunitinib, and thereby strengthen the identified efficacy contrasts seen between intracellular TKI and extracellular VEGF pathway inhibition in the neoadjuvant setting. In total, our manuscript now evaluates the use of 12 drugs. These include two VEGF RTKIs (sunitinib and axitinib); two neutralizing antibodies to VEGF (G6.31 and B20); one VEGFR-2 blocking adnectin (CT322), one neutralizing antibody to mouse VEGFR-2 (DC101); one vascular disrupting agent (OXi4503), one ALK/c-Met inhibitor (Crizotinib, PF1066), radiation (XRT); and three chemotherapeutics that include cyclophosphamide (CTX), a 5-fluorouracil oral pro-drug (UFT), and vinblastine (VBL) – the latter of which were administered either as a maximum tolerated dose (MTD) or as a low-dose metronomic (LDM) regimen. A total of 15 treatments were used in this manuscript either as single agents, combinations, or different doses.

This is a significant undertaking in terms of models used and drugs tested, and represents a highly unique and comprehensive comparative study into the effects of short-term treatment on localized and systemic disease.

3) Statistical significance: Several questions were raised by reviewers about statistical significance for survival studies following neoadjuvant treatment and surgical resection of an orthotopic tumor model. Specifically, there are three instances in the manuscript that were highlighted. These include A) the breast LM2-4 tumor model examining Sunitinib or OXi4503 treatment with different doses/schedules (Figure 2 – see Reviewer comments R1-3, R1-5, R2-7); B) the melanoma WM113/6 model examining Sunitinib, G6.31, B20, and CT322 (Figure 1 D-E and Figure 3C-D in original submission – see Reviewer comments R1-1, R1-2); and C) the melanoma WM113/6 model examining LDM/Sunitinib combinations (Figure 5 – see Reviewer comments R1-6, R1-8, and R2-10). Following a general comment below, we have discussed each of these instances and explained how we have altered the manuscript by adding data, analysis, and/or additional details. Where necessary, additional comments are mentioned in sections for each reviewer below.

General comments about statistical significance: We agree that conclusions of each study on survival must be based on statistical significance, however, we also agree with the Reviewer 2 (see comment R2-4) that survival ‘trends’ have value in certain instances, even if significance is not reached. Indeed, the use of highly complex models of surgical implantation and surgical resection may be such an instance where trends are important to consider. As in the clinical setting, spontaneous metastatic disease following removal of a primary tumor is highly variable and depends on a multitude of biological factors difficult to predict. Even in controlled animal studies (i.e., genetically identical mouse strains, standardized nutrition, living, mobility conditions, etc.) there is variability in disease progression rates and localization. This makes statistical powering of animal groups (we included between 6-15 animals per group in our studies) highly difficult, particularly if the treatment regimens studies are short-term. This disease variability underscores the importance of the extensive studies we undertook to optimize our chance of experimental and statistical success, namely, our determination of optimal surgical times (OSTs), etc., shown in Supplemental Figure S1. While we were able to modify the manuscript to address the reviewer concerns and support our claims with statistical significance, this was not possible in all instances (see 3C below). We hope that consideration of these details along with the noted changes will suffice for publication.

The changes made were the following:

A) Breast LM2-4 model (Sunitinib or OXi4503 treatment doses/schedules). It was noted by reviewers that the high dose treatment of Sunitinib or OXi4503 (see Figure 2F and Supplemental Figure S3F, respectively) showed trends of improved survival but these did not reach significance (P=0.094 and P=0.126, respectively). These values were derived from comparing treated groups to respective vehicle-treated controls. However, since the main purpose of this figure is to compare the two treated groups, we have now added an additional figure specifically comparing the post-surgical survival of these two treatment regimens (i.e., high/short vs. lower/longer dose). In both the Sunitinib and OXi4503 studies, this direct comparison produced a significant difference (P=0.009 and P=0.008, respectively) and is now included in both figures (Figure 2F and Supplemental Figure S3F, respectively). We have provided more justification for this comparison (particularly the similarities between vehicle-treated controls) in our response to Reviewer comment R2-7 below.

B) Melanoma WM113/6 model (comparison of Sunitinib, G6.31, and CT322): It was noted by reviewers that the neoadjuvant sunitinib treatment in the WM113/6 tumor model did not reach statistical significance either in the analysis of pre-surgical primary tumor growth or in the post-surgical survival (Figure 1D-E and Figure 3 C-D in original submission). This data remains in our revised manuscript; however we have altered both the description and the analysis used to explain this study. The specific changes we made are the following:

i) Figure 1D-E (original submission - now Supplemental Figure S2A-C): Since we have now added an additional syngeneic mouse model (RENCA) to Figure 1, we have moved the WM113/6 model to the supplemental section and altered our description in the results section to state that this model shows non-significant 'trends' rather than statistically significant changes. However, the results obtained with this model are still critical to highlight in the main text as they are supportive of our claim that pre-surgical neoadjuvant sunitinib treatment benefits do not necessarily predict for post-surgical benefits. We have rewritten the results section for Figure 1 to clarify this and to state that each of the three models (SN12, RENCA, and WM113/6) represent three different pre-surgical responses to sunitinib. This includes 1) a significant benefit (RENCA), 2) a non-significant benefit with 'trend' of improvement (WM113/6), and 3) a non-significant/non-trending benefit (SN12). Critically, each of these resulted in the opposite outcomes when compared to post-surgical survival. This includes no benefit (RENCA) or a *worsening* of survival (which is significant in SN12 model and a non-significant trend in WM113/6 model). Considering the complexity of the

three models used, and the supporting data shown in a fourth model (LM2-4, Figure 2), we feel there is justification for reaching the conclusion stated in the manuscript regarding a consistent lack of correlation between pre- and post-surgical outcomes following neoadjuvant VEGF TKI treatment (which now includes axitinib). In general, however, we agree that significant/non-significant results must be made clear and therefore the text in the results section for Figure 1 has been changed accordingly. More on this topic is highlighted in response to Reviewer 2 with additional data provided in attached Appendix (see R2-5, R2-6 and R2-11).

ii) Figure 3C-D (original submission - now replaced with new analysis in revised Figure 3C-D): In our original submission, Figure 3 demonstrated that there was a correlation between pre- and post-surgical responses to extracellular VEGF pathway inhibitors G6.31, B20, and CT322 but not to sunitinib. This was shown in the LM2-4 model (Figure 3A-B) and the WM113/6 model (Figure 3C-D in original submission). However, it was noted that the WM113/6 data did not reach significance (see Reviewer comment R2-8). We have made considerable changes to this figure to support the central claims, including new models, drugs, and analysis. These changes include: 1) we have added axitinib treatment to the LM2-4 model to Figure 3A-B. This figure now shows that neoadjuvant VEGF RTKI treatment with two drugs (sunitinib and axitinib) yield benefits in the pre-surgical setting but do not lead to benefits in post-surgical survival. This is in clear contrast to the B20, G6.31, and CT322 treatment that showed significant improvements in both settings. 2) We have removed the single analysis including only the melanoma model and added a novel figure that allows comparison of all models (where possible) to be analyzed together. For each model (LM2-4, SN12-PM6, WM113/L, and RENCA^{LUC+}), we have taken each treated group (in this case, sunitinib, axitinib, and B.20), and standardized to the average of their respective control groups. This yielded values for us to graph both pre- and post-surgical results following treatment and allowed us to pool multiple studies. In turn, this allowed graphing of `anti- or `pro-`effects on the primary tumor and metastasis to be shown simultaneously (See Figure 3C – top left panel). Such a comparison demonstrates that there is a clear (and statistically significant) difference in the neoadjuvant efficacy of the TKIs vs the B20 antibody, and strengthens the finding that the anti-primary and anti-metastatic properties for these treatments are not similar across multiple tumor models (Figure 3C). Furthermore, pooled pre- and post-surgical data analysed by Spearman rank correlation uncovered another interesting result, namely, that the magnitude of the primary tumor benefits (as compared to control) were

significantly correlated to overall outcomes in axitinib, sunitinib, and B20 treated animals (Figure 3D). This suggests that tumor size at time of resection following neoadjuvant treatment may indicate overall post-surgical benefits, independent of overall treatment benefit. Thus, the data from the WM113/6 studies remain in this figure but now are a part of a much larger analysis. The results for this section have been modified accordingly and include the following statement.

“Taken together, our results demonstrate that the pre-surgical efficacy of neoadjuvant therapy with an extracellular VEGF inhibitor on the primary tumor is more predictive of post-surgical survival outcomes than VEGFR TKI therapy and that the magnitude of tumor response after neoadjuvant therapy may be an independent surrogate marker of overall post-surgical benefits”

C) Melanoma WM113/6 model (Sunitinib and Low-dose metronomic therapy):

In Figure 5 we describe the use of neoadjuvant therapy to confirm the anti-metastatic properties of LDM CTX in combination with UFT (in the LM2-4 model) and VBL (in the WM113/6 model). The conclusions of this figure were that the anti-primary (but not anti-metastatic) sunitinib treatment could be combined with the LDM treatment which is anti-metastatic (but not anti-primary). Reviewers noted that this was only shown in the WM113/6 model where the combination did not reach statistical significance. Unfortunately, we were unable to modify these results with added experiments primarily because of the time needed to complete the work. First, the UFT compound was not immediately available for *in vivo* study within a timeframe amenable to resubmission, nor was it possible to repeat the studies in the WM113/6 in a reasonable timeframe (these studies can take up to 5 months or longer to complete because of tumor growth kinetics). Thus, while we have left Figure 5 as previously shown, we have included some modifications. First, we have modified the conclusions to clearly state that they are based on trends rather than significance. However, we have noted that the trends clearly show the *reversal* of the potentially negative effects of Sunitinib, with the ‘anti-metastatic’ effects of short-term neoadjuvant LDM CTX/VBL showing trends of *improved* survival and negating the (non-significant) sunitinib-induced pro-metastatic effect. Second, we have added a new analysis -- following the format introduced in Figure 3 – showing a summary comparison of the relative benefits of adding two treatments that may yield a net ‘anti-primary/anti-metastatic effect’. We hope the reviewers will agree that this remains pertinent to the overall discussion.

4) General manuscript improvements:

Reviewers and editor comments raised the need to modify the text and to condense the message. We have made several changes to the manuscript to improve the clarity, including completely rewritten or revised results and figure legends, as well as an expanded supplemental section with added experimental data and improved flow to the overall narrative. A list of these changes are as follows:

Changes made (by Figures)

Figure 1

A-D: unchanged

E-I: changed Melanoma model data moved to supplemental (Figure S2A-C), replaced with new (syngeneic) kidney tumor model. Added data– pre- and post-surgical BLI in Renca model following treatment

E-H: Added data– RENCA model (E), representative BLI images (F), BLI and tumor-bearing resected kidney data (G), overall post-postsurgical and post-treatment survival (H)

I: Added data – localization of spontaneous metastatic disease at endpoint for RENCA studies

Figure 2

A-B: Added Data: now includes more animals.

C: Added data: additional experiments to increase N.

D-F: unchanged

G: New figure: compares survival following short/high to low/long neoadjuvant Sunitinib

Figure 3

A-B: Added/moved Data: now includes Axitinib treatment, OXi4503 data moved to Supplemental Figure 3

C: New figure: Multi-model comparison of treatments with spearman rank correlation analysis.

D: New table: corresponding values for Figure 3C

Figure 4: unchanged

Figure 5:

A-D: unchanged

E: added new figure: Summary comparison of all treatments in format introduced in Figure 3B

Figure 6:

A: unchanged

B: added new data: Includes comparisons for RENCA metastasis model and axitinib treatment

Figure 7: moved to Supplemental Figure S7 (see comments made in response to R1-10).

Supplemental

Figure S1

A-Added Data: Increased N from 27 to 33

B-Unchanged

C-Added data: RENCA tumor model

D-data moved (was previously Figure S1C)

E-G-Graph improved: changed format and design for improved clarity

E-Added data: additional data from repeat experiment added

G-Added data: Renca model

H- Unchanged

Figure S2

A-C – added data - melanoma tumor model (previously Figure 1D-F)

Figure S3

Removed: Supplementary Figure 3 in original submission had included individual survival curves for some studies however this was removed because of general redundancy (data is shown in manuscript).

Added: A-G: Added summary of comparison of OXi4503 neoadjuvant treatment studies (previously these only partially shown in Figure 3 in manuscript). Figure S3G is new analysis.

Figure S4

Changed: previously Figure S5

Figure S5

Changed: previously Figure S6

Figure S6

Changed: previously was Figure S8

Figure S7

Moved/combined: previously Figure 7 in manuscript, now included in supplemental and combined with Figure S7 in original submission.

Figure S8

Changed/Added: Figure was previously Figure S2. It is now updated to include the RENCA model.

Referee #1 (Comments on Novelty/Model System):

The observations have only been demonstrated in SCID mouse models. If similar effects could be demonstrated in immune competent syngenic tumor models of metastatic disease this would further strengthen the conclusions.

Referee #1 (Remarks):

Summary:

In the manuscript EMM-2014-03989 "Neoadjuvant antiangiogenic therapy reveals contrasts in primary and metastatic efficacy" Ebos and co-workers investigate the potential of modeling the neoadjuvant application of different types and combinations of antiangiogenic and chemotherapy in three previously established postsurgical orthotopic mouse models of advanced metastatic disease. The authors develop the necessary experimental conditions for effective neoadjuvant treatment and surgical resection to ensure metastatic potential as a prerequisite for their subsequent assessment of response on primary tumor growth and postoperative metastatic recurrence. They conclude, that preclinical neoadjuvant tumor models can be used to differentiate between anti-primary and anti-metastatic treatment effects that could lead to rational combination strategies to improve perioperative outcomes in patients.

General judgment:

Overall, the manuscript is addressing the very interesting and clinically important problem of frequently discrepant therapeutic outcomes observed between preclinical and clinical studies. The frequent failure to predict successful therapeutic strategies for clinical trials on the basis of preclinical efficacy is partly due to the lack of adequate models of advanced metastatic disease given that traditionally most preclinical tumor models only assess primary tumor growth response. The problem has prominently been illustrated by the disappointing results of sunitinib in clinical trials. The authors add to this field by using their unique platform of recently established postsurgical tumor models with spontaneous metastatic disease, to explore the effects of different VEGF pathway inhibitors with and without chemotherapy in the neoadjuvant setting. They show that counter-intuitively the presurgical benefit of neoadjuvant therapy does not consistently predict for postsurgical disease recurrence and survival, which is influenced by a plethora of variables, which this study begins to shed light on. While still relatively preliminary, these studies could potentially offer new insights into the biological differences between primary and metastatic disease and help explain their differential response to therapy.

While the concept of the manuscript is absolutely worthy, the data in the manuscript clearly illustrate the huge obstacles involved in developing appropriate tools to predict outcomes for patients receiving neoadjuvant antiangiogenic therapy. The authors approach is based on a long track record of studying the effects of metronomic chemotherapy and antiangiogenic therapy in various tumor models. A weakness of the manuscript is that the observations have only been demonstrated in SCID mouse models. If similar effects could be demonstrated in immune competent syngenic tumor models of metastatic disease this would further strengthen the conclusions. Also, the translational potential for predicting drug combinations that have an increased likelihood of success in clinical trials is not immediately obvious. If anything, the results clearly show that even for a given cancer model the outcome can be different for primary and metastatic tumors, even when the same treatment is used in different concentrations or application schedules. Whether these data translate to other tumor models or into the clinical setting is uncertain. Regardless of this, there is clear value in this approach in terms of developing and testing a preclinical platform to explore drug regimes in the neoadjuvant setting. These findings should therefore be viewed as a hypothesis generating starting point for further investigations that compare specific drug classes or dosing schedules in order to advance our understanding of the underlying cancer biology and the broader principles of metastatic progression that may eventually be leading to future clinical applications.

Specific Points- Major:

R1-1. Fig. 1E and F: The authors state on Page 7 that there is a reduction in WM113/6-4L tumor volume and tumor weight after 14 d compared to vehicle Fig. 1E/G. There is no statistics to support this and the data does not seem to show any relevant differences (which would fit with the data presented for the RCC model in Fig. 1A/B). Could the authors clarify this?

Please see our General Comments (#3Bi) for a comprehensive explanation of the changes made regarding this data. Briefly, this experimental data has been moved to supplemental Figure 2 (replaced by the RENCA model) and the text has been altered to highlight noted trends, indicating non-statistical significance.

R1-2. Since Fig. 1G does not reach significance. The general conclusion concerning both kidney and melanoma models is somewhat misleading. In any case would the final conclusion that primary tumor response following neoadjuvant sunitinib therapy failed to predict post-surgical survival not be better supported by the breast cancer model, since Fig. 2A-C show a reduction in primary tumor growth in response to sunitinib treatment that again fails to translate into a postsurgical survival benefit similar to the RCC?

As per the above statement, we have addressed this in our general comments (#3Bi and 3Bii), as well as altered the text to clarify that, in four models of metastasis (including those depicted for breast in Figure 2A-D), a two week neoadjuvant treatment with sunitinib yielded primary tumor effects that were not predictive of postsurgical outcomes. In our discussion (and listed comments), we have clarified why these studies (including those not reaching statistical significance, such as in melanoma), represents a consistent finding with this treatment regimen.

R1-3. Fig. 2F: the conclusion that short term high dose sunitinib may offer improved postsurgical outcomes compared to longer-term, lower dose treatment is intriguing but does not appear to be statistically supported by the data. This is unfortunate, as this could be easily translated into clinical trials.

We have detailed the modifications to this section of the manuscript in two separate places, including in our General Comments (#3A) and in comments made to reviewer 2 (see R2-7 below). We hope that the changes made including new analysis, text, and supplemental figures will be acceptable.

R1-4. Concerning the metastatic breast CA model (LM2-4 Luc+): in a recent Cancer Research publication by Guerin et al., 2013 the Kerbel group commented on the use of this tumor cell line: "We did not use a luciferase-tagged clone of LM2.4 (19) because we have found that these cells have a reduced ability for spontaneous metastasis (unpublished observations), the basis of which is currently unknown." Could the authors comment on this?

We thank the reviewers for raising this very important point. The sentence in the Guerin *et al* paper is technically correct but we acknowledge that it is somewhat misleading. In this instance, the authors were explaining that the use of the untagged-LM2-4 cell line was dictated by the fact that the post-surgical disease progression was the more characterized of the two cell lines at the time, and therefore was the best choice for use in their paper, which focused on treating post-surgical disease. Both the tagged and untagged variants metastasize, but it appears that this happens at different rates. Importantly, the luciferase-tagged LM2.4 clone has been validated several times in the past (Ebos et al, 2008; Ebos et al, 2009), and the studies described in this manuscript represent the most comprehensive study of the parameters and variables that influence metastatic potential (for either tagged or untagged LM2-4 variants). As is shown in Supplemental Figure S1A and S1E, we used data from 33 animals (4 separate experiments) to identify an optimal surgical time frame in which to surgically excise tumors to produce post-surgical spontaneous metastatic disease. This was necessary to establish a pre-surgical window long enough to conduct neoadjuvant studies. To further support the reliability of the metastatic potential, we have provided a summary of all of our experiments that show the post-surgical survival

of neoadjuvant vehicle-treated mice implanted with the luciferase-tagged LM2-4 cell line, which were then resected (see Appendix A1A). This data represents 58 mice and 8 different experiments. As can be seen, the surgical time-point clearly impacts the overall survival (discussed in more detail in responses to R1-11 and R2-7 below).

R1-5. Fig. 3 and S3E: Again the statement that neoadjuvant OXi4503 efficacy was 'significantly' improved by short term, high dose treatment is not statistically supported by the data in Fig. S3E.

We have addressed the topic of the experiments with OXi4503 in our General Comments (see comment #3A) and have mentioned several details about the importance of these studies in response to Reviewer 2 (R2-7 below). To summarize, we have modified the format of the studies comparing the two neoadjuvant regimens involving OXi4503 and included in Supplemental Figure S3. We have also added a graph directly comparing the post-surgical survival of the two treatments, which achieves statistical significance. We have made more detailed comments to justify these comparisons in response to R2-7 below. In the text, Supplemental Figure S3 is now referenced as follows:

“Shorter (7 days) higher dose (120mg/kg/day) neoadjuvant sunitinib treatment showed significantly improved survival compared to sunitinib administered in lower doses over a longer period (60 mg/kg over 14 days, respectively). Interestingly, similar observations were made in the same model with a vascular disrupting agent, OXi4503, given neoadjuvantly at higher doses (50mg/kg) once 7 days providing a significant survival advantage over a lower dose (10mg/kg) given twice in 14 days. These postsurgical differences contrasted with the significant benefits observed in the presurgical setting following neoadjuvant therapy (Supplemental Figure S3A-S3G)”

R1-6. Fig. 5D: The authors acknowledge that the data in Fig. 5D is not statistically significant. The statements made in the heading of this paragraph and the title of the figure are not statistically supported by the data shown.

We have addressed this comment directly in our General Comments (see 3C) because it was raised by reviewers here and elsewhere (see R1-8 and R2-10 where we have included some additional comments). In short, we have modified the text to better reflect the results of the study and included an additional type of analysis (see Figure 5E) to highlight the changes seen in the combination-treated

group. We hope this, as well as the extensive explanations in other sections, will suffice.

R1-7. The authors suggest that the Neoadjuvant Efficacy Score (NES) calculated in Fig. 6B might serve as a predictor of anti-primary or anti-metastatic efficacy and that it could serve as a tool to compare treatments and predict drug combination strategies to improve overall outcome. However, the predictive value of the NES is not obvious, given that the same treatments can have different effects in different tumor models, or depending on dose. Please clarify.

We thank the reviewer for raising this point as we agree that it is somewhat difficult to derive the predictive value of our combined NES calculations upon initial review. This, in part, could stem from the fact that we have sorted the table based on the value of the NES, rather than either by the primary tumor size or the overall survival. We did this because the main point of the NES is to use these scores to determine the best neoadjuvant therapy (i.e., the better NES score, the better the overall pre- and post-surgical effects of neoadjuvant treatment). However, by rearranging the graph and sorting by the primary tumor values, the disparity of the anti-primary (but not anti-metastatic) and the anti-metastatic (but not anti-primary) become more clear. We have included an example of how the data would look if re-sorted using only the LM2-4 studies in Figure 6 (see Appendix A2A). By sorting according to primary tumor benefit the differential efficacies of the sunitinib and LDM CTX/UFT treatments are clearly shown at opposite ends in terms of efficacy. While we do think that presenting the graph in this manner would improve the clarity to readers for determining potential combinations, we have left the graph as it was (i.e., sorted by NES score) as this fits best with the manuscript.

R1-8. The data summarized in Fig. 6B is clearly useful for the purpose of comparing the therapeutic effects and identifying potentially synergistic drug combinations. In particular, the concept would be strengthened if rational combinations for pre- and postsurgical therapy could be predicted on this basis. For example if differential efficacy of high and low dose OXi4503 or Sunitinib could be recapitulated in one of the other tumor models or if Sunitinib low (60mg), which is effective in reducing primary tumor growth could be combined successfully with post-surgically active drugs such as OXi4503 or LDM CTX/UFT.

We thank the reviewers for this comment and we agree that additional experiments combining two treatments with differential pre- and post-surgical efficacies would represent the ideal use for the NES methodology. As a proof of principal, this manuscript has done this, though we have already noted that the combination studies with sunitinib and CTX/VBL did not reach statistical significance. We have detailed this in our General Comment (3C), along with an explanation for why it was not feasible to conduct further studies to identify more combinations based on the NES. Current studies are underway to examine this

with several different treatments using multiple models but this is currently beyond the scope of this manuscript. We hope the reviewers will agree with this as well as our previous comments (and decisions) on this model.

R1-9. A general weakness of the manuscript is that the observations have only been demonstrated in SCID mouse models. If similar effects could be demonstrated in immune competent syngenic tumor models of metastatic disease this would further strengthen the conclusions.

As we have outlined in our General Comments section (See comment #1) we have now included a mouse kidney tumor cell line (RENCA^{LUC+}) in a syngenic model of metastasis (included in Figure 1, Figure 3C, Figure 6B, and supplemental Figures S1C, S1G, and S8B).

R1-10. The results in Fig. 7 should be discussed some more.

Following the comment by reviewer 2, who suggested moving this figure to the supplemental materials (see R2-12) and the editors' suggestion to shorten and streamline the manuscript; we have moved this data to Supplemental Figure S7 and provided additional details in the supplemental results and methods sections, and added some comments to the Discussion section. Though we feel this is the correct decision, it is difficult nonetheless for several reasons. First, the experimental methodology involved in generating, staining, and analyzing the Tumor MicroArray created for this study (with over 100 tumors) represents a significant undertaking. Second, the results are interesting and do suggest that Ki67 may have potential utility in predicting post-surgical benefits following neoadjuvant therapy. To expand on this latter point, the purpose of this figure is to quantify expression levels of markers of Ki67, CD31, and vimentin in resected primary tumors (WM113/6) following neoadjuvant therapy and compare with post-surgical overall survival. While the results do suggest that elevated Ki67 levels can predict for increased survival following treatment with sunitinib, it is interesting to note that the opposite trend was observed following neoadjuvant B20 and CT322 therapy (i.e., elevated Ki67 in tumors correlated with decreased survival) (see Supplemental Figure S7C). Therefore Ki67 may represent a potential biomarker for neoadjuvant therapy which may allow stratification of patients in terms of post-surgical risk, however, this may depend on the type of inhibitor used. Regardless, we have agreed to move this information to the supplemental because the WM113/6 model experiments on which this data is based has now been moved to the supplemental (See our General comments, 3Bi), in part, because the overall survival in these studies did not reach statistical significance. Future studies will examine whether these and other molecular markers change in response to neoadjuvant therapy and test whether they may have utility as biomarkers to predict post-surgical benefit when survival is significantly improved.

R1-11. In Fig. S1 A and B it is not clear to me which tumors belong to which primary surgery time point and why the minimum OST threshold was

determined at different time points for the different tumor entities. What is the value of the linear regression lines?

The reviewer has raised a very good point about the variables we chose to control for in our studies to optimize the evaluation of neoadjuvant therapy. To clarify, in Supplemental Figures S1A-S1C, we examine two primary tumor variables which may influence the metastatic potential following surgery (as determined by overall survival following primary tumor removal). These are tumor size and time of surgery. For the tumor size, we pooled tumor measurements (weights, BLI, or organ weight, depending on the model) and compared with survival. For the time of surgery (which we used to determine a range of optimal surgical times, or OST), we looked at two parameters, namely how many mice reached the end of the experiment without postsurgical recurrence (i.e., surgery was 'curative'), and how many had metastasis (as measured by BLI at our designated threshold), which we then used as a gauge of residual cancer burden (RCB). The reviewer is correct that we did not also show the relationship between each of these surgical timepoints and survival. Therefore, we have compiled this information for the LM2-4^{LUC+} cells in Appendix Figure A1A for reviewer consideration as an example of what this would look like. As expected, the earlier the surgery, the more mice were surgically cured. What is interesting is that there seems to be a timepoint where the surgical time does not seem to influence the overall survival (i.e., the mice succumb to the disease at the same rate). While our study is aimed to show that there is an upper limit to when surgery can take place (i.e., the aforementioned RCB), Appendix Figure A1A suggests that there is a threshold for which the primary tumor can impact the 'metastatic-ness' in terms of eventual disease progression. Unfortunately the implications for these results extend beyond the scope of this manuscript and we have recently begun a work to use mathematical modeling to explore this issue in more detail. The linear regression lines have been added here to show the correlation between the two variables and to demonstrate that there are clear statistical correlations between pre-surgical tumor burden and post-surgical survival (shown in 3 of the 4 models).

R1-12. While the manuscript is generally well written, it could benefit from a more focused comparison of equivalent results between tumor models and a more critical assessment of the data obtained in different tumor models. The figures would benefit if direct comparisons between equivalent data would be made easier, for example by naming the tumor models at the top of each figure and arranging the corresponding data below for each model.

We thank the reviewer for this suggestion. For each model, in the figures and supplemental figures, we have now added a clear description of what cell line was used. We agree this improves the clarity of the paper.

Specific points - minor:

1. Page 10: improved postsurgical survival after short term high dose OXi4503 is documented in Fig. S3E (not D).

-this has been corrected

2. Fig. 6B: OXi4503 should be 50mg/kg

-this has been corrected

3. Page 14: "Similar correlations with CD32 were not observed following in SN12-PM6 Luc+..."

-this has been corrected

Referee #2 (Comments on Novelty/Model System):

See detailed comments to author

Referee #2 (Remarks):

In this manuscript, Ebos, Kerbel and colleagues employ three mouse models of orthotopic and metastatic human cancer (breast, kidney and melanoma) to comprehensively study the effects of neoadjuvant therapy on (i) primary tumor growth; (ii) post-surgery survival and metastatic burden. They employ a large number of therapeutics, primarily antiangiogenic drugs such a broad-spectrum TKI (sunitinib), a vascular disrupting agent (VDA), and VEGF/R2 blocking antibodies/adnectin. Based on their findings, the authors conclude that post-surgical survival does not consistently correlate with primary tumor responses in the neoadjuvant setting.

The study is of significant interest as it represents an attempt to comprehensively study the effects of distinct types of neoadjuvant treatments - mostly antiangiogenic - on post-surgery survival in mouse models of metastatic cancer. While this study certainly merits publication, I feel that the clinical relevance of some of the models/regimes is questionable and should be clarified. Furthermore, some of the authors' conclusions are at times at odds with the data presented. The authors may consider the following comments and criticisms to clarify some important aspects of their study.

General criticisms

R2-1. Clinical relevance. The authors used 3 aggressive tumor models, which readily metastasize to multiple organs. The choice to perform neoadjuvant therapy followed by a post-surgery observation time is not completely justified. Because the tumors already metastasize during the treatment, it would be interesting to see whether post-surgery adjuvant therapy (same or another drug) would reverse the prometastatic effects seen with certain drugs, or generally improve survival. In a clinical setting, these primary but already metastatic tumors would be treated with a short neoadjuvant treatment (if any) followed by (cycles of) adjuvant therapy.

The reviewer raises a critical point about the study of neoadjuvant therapy preclinically, along with many of the associated challenges. First, however, it should be noted that pre-surgical treatment followed by post-surgical observation is conducted clinically and is standard practice for neoadjuvant studies. However, as the reviewer states, the key point here is whether the human (or the mouse in this instance) is considered 'tumor free' following both neoadjuvant treatment and surgery. The challenge is that the designation of 'tumor free' is defined by

pathological parameters that constantly change, not only over time because of technological advances in residual disease detection, but also because of institutional or national standard differences (this was recently reviewed in (Fumagalli et al, 2012). In Supplemental Figure 1 and in the Supplemental results section, we have outlined some examples of the disease progression we encountered in all of our models at time of surgery. As stated in the methods section, if a mouse had obvious visual spread of tumor (classified as non-localized) they were removed from the study and sacrificed. However, for others where nothing was visible, there were some animals that did show distant or local disease by BLI (examples shown in Supplemental Figure S1H). In this instance, mice were permitted to remain in the study because the sensitivity of BLI cell detection exceeds any clinical detection level for micrometastatic disease used in humans, something that underscores the limitations of what 'tumor free' designations are clinically. Thus while there is a chance that some of our mice were treated synchronously (i.e., they had both primary and metastatic lesions), our visual inspection during surgery did not allow confirmation. In our opinion, we feel that this inclusion/exclusion criteria represents the best chance of recapitulating the clinical setting and examining neoadjuvant therapy. It should be noted that we are not aware of any previous studies that have attempted to study neoadjuvant therapy with similar scrutiny of how these parameters could affect outcome. This is likely because of the extreme difficulty (and high cost) of undertaking such surgical models as well as the difficulty in generating tumor cell lines that readily metastasize. As we state in our extensive supplemental materials, the challenge is finding the optimal surgical and treatment time that allow for tumors to be metastatic (unlike a clinical neoadjuvant trial, surgically 'cured' animals would be of little use for us experimentally) yet have not been obviously growing systemically (thus compromising the stated goals of differentiating between primary and metastatic treatment effects). As this manuscript clearly shows, this is extremely difficult in practice. However, we feel this study will assist in setting a guideline for more preclinical studies in the future. We hope the reviewers will agree.

R2-2. CTX is not the preferred cytotoxic drug for neoadjuvant therapy in breast cancer. A combination of an anthracycline and a taxane would have been most relevant. Although I understand that this study is primarily focused on antiangiogenic drugs, the authors should comment on this point or test the most relevant drugs in a side-by-side comparison.

We thank the reviewers for this comment and agree that the MTD CTX is not the most clinically relevant choice. Though we would agree that the most appropriate choice of treatment would be the suggested combinations, we have not performed additional experiments primarily because the MTD CTX neoadjuvant treatment in our model was included as a proof-of-principal study to show that chemotherapy treatment can yield pre-surgical benefits that translate into post-surgical benefits (as seen in patients) and to serve as a contrast to the sunitinib treatment effects in the same model. In essence, we wanted to show that pre- and post-surgical

benefits were *possible*, and therefore we used the MTD CTX. Because of this, we feel the addition of additional chemo combinations would not significantly add to this figure. We hope the reviewers will agree.

R2-3. The clinical relevance of neoadjuvant therapy for metastatic melanoma is unclear. The authors should elaborate on this.

Similar to the statement above, while the reviewers are correct in noting that neoadjuvant antiangiogenic treatment is not currently an option for melanoma, we have included it in our studies as an additional example of a metastatic model where the pre- and post-surgical effects of treatment can be studied. In this particular model, the cell variants were derived previously for high metastatic potential following multiple *in vivo* selections (Cruz-Munoz et al, 2008). Thus, because of the difficulty in obtaining such metastatic cell lines, and in addition to the (relative) surgical ease of tumor resection (this model rarely has localized invasion or current metastatic growth), we chose to include it.

R2-4. Some of the results are commented based on data that failed statistical analysis. While I am convinced that "trends" toward differences are worth reporting, the strength of the conclusions is certainly undermined in such cases.

We thank the reviewer for this comment. We have addressed this and other concerns about the statistical basis for conclusion in our experiments in detail in our General Comments (see comment 3).

R2-5. It seems that the authors overemphasized in the Abstract and elsewhere in the ms the discrepancies between pathological responses in the primary tumors and mouse survival. Overall, the data seem to show the opposite, i.e., a general correlation between the two (see Figure 6). The authors should revise their statements to more rigorously reflect this fact.

We appreciate these comments by the reviewer and would like to note that this concern is also raised in subsequent comments (see R2-6 and R2-11), so we have tried to address all in this section, and in our General Comments (see 3Bi). We have tried to detail our justification for the claim that VEGF RTKIs sunitinib (and axitinib) show a lack of correlation between the primary tumor response following neoadjuvant treatment and the post-surgical survival. Further to the points already raised, we hope that the inclusion of additional analysis in Figure 3A-D of the manuscript will support the conclusion that the predictive value of primary tumor responses is not consistent for eventual metastatic recurrence and overall survival. What we have stressed in our study (including our revised results section for Figures 1-3), is that 14 day neoadjuvant treatment with sunitinib or axitinib showed differences in the pre- and post-surgical setting and, in all

instances, these differences were worse than the primary tumor activity would have predicted (see our General Comments 3Bii where this is explained in detail). Also, please note the comments made in our response to R1-7 above, particularly the inclusion of our modified Figure 6 in Appendix A2A, which highlights more clearly the contrast between pre- and post-surgical responses for each therapy. We hope the reviewer will agree with this conclusion and the presentation of the results as they stand.

R2-6: Figure 1. At the end of the first section of the Results, the authors state that "primary tumor response following neoadjuvant therapy DID NOT predict for benefit in post-surgical survival...". However, in both kidney and melanoma models sunitinib (60mg/Kg) achieved neither complete nor partial pathological responses as primary endpoint. So, it is somewhat expected that post-surgical survival be not increased by neoadjuvant treatment in these models. If any, post-surgical survival is slightly decreased in the kidney cancer but not melanoma model (KM, p value is 0.97). A more appropriate conclusion for this first set of experiments should therefore be "LACK OF primary tumor response following neoadjuvant therapy DID predict for LACK OF benefit in post-surgical survival...". Note that panels G and H are incorrectly called in the main text (there is no panel I).

Please refer to our response to R2-5 for this section.

R2-7: Figure 2. These experiments are, in principle, interesting, particularly the comparison between the 14-day long neoadjuvant sunitinib (60mg/Kg) and the "condensed" 7-day long neoadjuvant sunitinib (120mg/Kg). However, it is somewhat unfortunate that, in the condensed setting, tumors were surgically removed one week earlier than in the other experiment. Is there a specific reason for selecting such schedule rather than a neoadjuvant therapy -> drug withdrawal -> surgery schedule that would overlap with the 14-day schedule? The interpretation of the results may therefore be confounded by this factor, and a direct comparison between the two schedules seems not possible. Also, the improved post-surgery survival in the condensed setting is questionable (panel F). It would be informative to directly compare the low and high dose sunitinib treatment in a single experiment and reassess primary endpoints as well as survival benefits.

The reviewer has raised critical points regarding the impact of short-term/high-dose vs. longer-term/lower-doses neoadjuvant treatment and the experimental methodology chosen (including the use of 'gaps' treatment schedule). Prior to initiating these comparative studies, we weighed several experimental variables that needed to be controlled. These variables can have a significant impact interpreting the effects of neoadjuvant therapy pre- and post-surgical disease progression (a topic we have detailed in Supplemental Figure S1 and in the Methods section). For studies with Sunitinib and OXi4503 (Figure 2 and

Supplemental Figure S3, respectively) we considered the following variables as critical for the comparisons of different doses and resection times. These were:

1. The day cells are implanted: having two experiments start on the same day allows for age-matched mice from the same vendor to be included.
2. The cell preparation and implantation procedure: There is always some variability in cell preparation and passage number. Having all animals implanted at the same time from the same preparation ensures variability is minimized when comparing treatments.
3. The tumor size at resection: Supplemental Figure S1 shows that tumors of different size have different impacts on overall survival.
4. The time between implantation and resection (tumor burden duration): Depending on the kinetics of tumor growth, the longer exposure a mouse has to a tumor may increase the potential for metastatic spread (more details below).

In Figure 2 and Supplemental Figure S3, we controlled for variables #1 and #2 by initiating the experiments together on the same day. Though this is not shown directly in our manuscript because we represented the values of the resected tumors as standardized to control (see Figure 2B and 2E), the tumor sizes in the vehicle-treated controls are nearly identical, despite the different resection days. In Appendix Figure A3A, we have included these raw tumor weight values showing they are of equivalent size at time of resection (as are the sunitinib-treated tumors). This means we have controlled for variable #3 as well. However, by choosing to control for #1-3, we were not able to control for variable #4. Importantly, and as we stated in our response to R1-4, Appendix Figure A1A shows that the impact of resection date on post-surgical survival seems to have a threshold in terms of how it influences metastatic potential. Put simply, this means that, at a certain time-point, mice essentially succumb to disease at the same time, regardless of surgery time. In confirmation of this, Appendix Figure A3B shows that the vehicle-treated control mice from Figure 2B and 2E have nearly the identical post-surgical survival, despite resection times 7 days apart. Thus, it appears that that not controlling for #4 above (which we view as mutually exclusive to #1-3), had the least impact on the outcome of the study.

As the reviewer suggested, another option in such neoadjuvant comparisons could include controlling for all of these variables by having a 'gap' in treatment. We feel this is not an option for several reasons. First and foremost, there is clinical evidence that stopping therapy may have an impact on rebounds in tumor growth – which in turn could affect post-surgical outcomes. Though this topic remains debated in the field of antiangiogenic therapy both by us and others (Blagoev et al, 2013; Ebos & Kerbel, 2011), the impact of stopping therapy in the neoadjuvant setting may influence overall outcomes. A recent retrospective analysis of data from a RCC patients receiving presurgical antiangiogenic therapy suggested the potential for an endothelial proliferative rebound following break periods, and this was raised as potentially having a potentially negative impact on overall postsurgical disease progression (Ebos & Pili, 2012). In the discussion of our paper we have the following paragraph on this topic (with citations included):

“Our results showing increased dose and shortened surgical window overcoming putative negative (or negligible) post-surgical impact on overall survival could warrant consideration in clinical neoadjuvant trials with VEGFR TKIs, where parameters of tumor dosing and tumor size are still being investigated in terms of assessing overall benefit (Kroon et al, 2013). Already evidence from retrospective studies investigating pre-surgical cytoreductive sunitinib treatment in RCC suggest that parameters of treatment stage (Bex et al, 2011) and primary tumor reduction (Abel et al, 2011) may play a significant role in patient outcomes. In this regard, our results demonstrating that the magnitude of primary tumor response following neoadjuvant therapy correlates with overall survival could support these findings. Furthermore, it is also possible that alterations in standard pre-surgical dosing could alleviate concerns about potential break periods, or gaps in treatment, that typically occur in patients receiving neoadjuvant therapy (i.e., toxicity). Related to this, recent retrospective studies in RCC patients receiving pre-surgical VEGFR TKIs showed an increase in proliferative tumor endothelial cells (ECs) in patients that had a longer treatment break before surgery (Ebos & Pili, 2012; Griffioen et al, 2012). But the same study showed that bevacizumab did not yield similar elevations in proliferating ECs. This could be potentially attributable to the aforementioned longer half-life of antibodies compared to TKIs - but it also may be consistent with our preclinical neoadjuvant studies. Future studies will address whether treatment gaps can influence post-surgical survival and/or metastatic disease distribution in the neoadjuvant setting.”

Thus adding such ‘gaps’ in our studies was not included because it extends beyond the scope of our current studies. However, it is a critical topic and for this reason we have dedicated separate studies evaluating the impact of gaps in treatment in the neoadjuvant setting and are preparing a manuscript for publication.

R2-8: Figure 3. The breast cancer data look compelling. I would add the high VDA dose in Figure 3A-B and move the melanoma data (C,D) to the supplemental data.

We have addressed the inclusion of the studies with the OXi4503 compounds in our General Comments (see comment #3A) and in our response to reviewer comments (See R1-5 and R2-7). While we agree that the OXi4503 data is

compelling and highly supportive of the observations made with sunitinib treatments in the LM2-4 breast model comparing two treatment regimens, however, we have opted to include these studies only as supplemental information so as to streamline and shorten the overall manuscript length. We hope the reviewers will agree with this decision.

R2-9: Figure 4. In their interpretation of the results from this set of experiments (7-day pretreatment schedule followed by iv injection of breast or melanoma cancer cells), the authors should more rigorously consider the significance of the statistical analysis of the data. Indeed, no drug provided survival benefit as monotherapy, contrary to the message conveyed by the authors. Interestingly, aggressive treatments like high dose sunitinib, MTD CTX and RT worsened survival. I believe that this is the main message of this set of data. Furthermore, the clinical significance of the 7-day preconditioning schedule is unclear and should be discussed more thoroughly. Does this mimic a post-surgery adjuvant setting?

The inclusion of the pre-conditioning experiments follow the studies we previously reported in 2009 (Ebos et al, 2009), which demonstrated that sunitinib-pretreatment (120 mg/kg/day) could accelerate metastatic disease when given only for 7 days prior to i.v. tumor cell inoculation. We have outlined the impact of this paper (and resulting papers that followed it) in our discussion section, along with associated references. The inclusion of these pretreatment studies in this manuscript represent an extension of these previous studies, and our hypothesis that such off-target host responses to therapy may have the greatest clinical impact on neoadjuvant treatment, as this setting typically involves a patient with a localized tumor receiving systemic treatment (where host responses may have an impact). We have included the following sentence in our discussion.

For our studies, we chose to evaluate the neoadjuvant setting to determine whether these putative ‘pro-metastatic’ treatment effects could be observed in a clinically relevant model. In this regard, neoadjuvant therapy could potentially allow for testing of off-target ‘host’ effects (since it involves systemic treatment) and allow differentiation between primary tumor responses and post-surgical disease recurrence following treatment cessation.

Our inclusion of multiple treatments, including RTKIs, chemotherapy and multiple extra-cellular VEGF pathway inhibitors was to serve as a starting point for eventual comparison (or explanation) for potential differences in the neoadjuvant setting. As our Figure 4 shows, the lack of pro-metastatic effects in the pre-conditioning setting using the VEGF pathway inhibitors (compared to the VEGF RTKI, sunitinib) could theoretically explain the post-surgical differences we observed with these treatments (detailed in Figure 3). It is interesting to note that such metastatic host-responses seem not to explain why LDM combination

therapy seems to have an anti-metastatic effect, and suggest that this treatment is preventing some property of the localized primary tumor, rather than altering the systemic host environment.

R2-10: Figure 5. These data essentially reproduce published work from these authors that low-dose, metronomic cytotoxic therapy (LDM, here CTX/5FU) improves post-surgery mouse survival but does not induce pathological responses in the primary tumor in a breast cancer model. Panels A-B show the breast cancer model, panels C-D the melanoma model. Only in the latter model was LDM combined with low-dose sunitinib to explore combinatorial effects on the primary tumor and post-surgical survival. However, the data do not show statistically significant differences, so the message of this combination experiment is unclear. It would be very interesting to see the effects of sunitinib + LDM in the breast cancer model, where sunitinib monotherapy is effective on the primary tumor but does not improve survival (Figure 2B-C).

We thank the reviewer for this comment and we agree that such studies would ideally be added to this manuscript. However, such experiments have not been added for the detailed reasons we listed in our General Comments (see comment 3C). This topic was mentioned in Reviewer responses R1-6 and R1-8, where we have provided some additional explanation. We hope the reviewers will agree with our decision and the changes to the text that were made.

R2-11: Figure 6. This Figure provides an interesting and comprehensive summary of the results. It is clear from this beautiful representation of the results that in most instances the data suggest concordant responses in primary versus metastatic tumors (i.e., effects on primary tumors and mouse survival). LDM and high-dose sunitinib represent notable exceptions, but apparently only in the melanoma model. Based on these findings, the authors should revise their claim that the benefits observed in primary tumors often do not correlate with overall survival. In general, the data seem to show the opposite.

Please refer to our response to R2-5 for this section.

R2-12: Figure 7. This Figure is a bit off-topic and may be move to the supplemental data.

We have moved this data to Supplemental Figure S7 as suggested, and provided a detailed explanation about its findings in our response to Reviewer 1 (see R1-10).

Referee #3 (Remarks):

This is an excellent paper that addresses important questions related to neoadjuvant antiangiogenic therapy. By using various models of spontaneous metastasis, the authors have evaluated pre-surgical therapy with regard to primary tumor response and metastasis. They show that primary tumor response and overall survival sometimes do not correlate, and that significant differences exist between different classes of VEGF inhibitors. The results obtained using different therapeutical regimens are complex, hence the authors established a "neoadjuvant efficacy score" which may be helpful to evaluate the efficacy of neoadjuvant therapy. Overall, the paper is an important contribution that may eventually help to improve the design clinical trials.

The preclinical models used in this study are well suited to address the efficacy of neoadjuvant therapy, the results are convincing and well presented. However, the following point needs to be addressed:

- on page 7, the numbering of Figure 1 is partially confused: Figure 1H shows the evaluation of the metastatic sites, and Figure 1I does not exist.

-this has been corrected

Reference List

- Abel EJ, Culp SH, Tannir NM, Tamboli P, Matin SF, Wood CG (2011) Early primary tumor size reduction is an independent predictor of improved overall survival in metastatic renal cell carcinoma patients treated with sunitinib. *Eur Urol* 60: 1273-1279
- Bex A, Blank C, Meinhardt W, van Tinteren H, Horenblas S, Haanen J (2011) A phase II study of presurgical sunitinib in patients with metastatic clear-cell renal carcinoma and the primary tumor in situ. *Urology* 78: 832-837
- Blagoev KB, Wilkerson J, Stein WD, Motzer RJ, Bates SE, Fojo AT (2013) Sunitinib does not accelerate tumor growth in patients with metastatic renal cell carcinoma. *Cell Rep* 3: 277-281
- Cruz-Munoz W, Man S, Xu P, Kerbel RS (2008) Development of a preclinical model of spontaneous human melanoma central nervous system metastasis. *Cancer Res* 68: 4500-4505
- Ebos JM, Kerbel RS (2011) Antiangiogenic therapy: impact on invasion, disease progression, and metastasis. *Nat Rev Clin Oncol* 8: 210-221
- Ebos JM, Lee CR, Bogdanovic E, Alami J, Van Slyke P, Francia G, Xu P, Mutsaers AJ, Dumont DJ, Kerbel RS (2008) Vascular endothelial growth factor-mediated decrease in plasma soluble vascular endothelial growth factor receptor-2 levels as a surrogate biomarker for tumor growth. *Cancer Res* 68: 521-529
- Ebos JM, Lee CR, Cruz-Munoz W, Bjarnason GA, Christensen JG, Kerbel RS (2009) Accelerated metastasis after short-term treatment with a potent inhibitor of tumor angiogenesis. *Cancer Cell* 15: 232-239
- Ebos JM, Pili R (2012) Mind the Gap: Potential for Rebounds during Antiangiogenic Treatment Breaks. *Clin Cancer Res* 18: 3719-3721
- Fumagalli D, Bedard PL, Nahleh Z, Michiels S, Sotiriou C, Loi S, Sparano JA, Ellis M, Hylton N, Zujewski JA et al (2012) A common language in neoadjuvant breast cancer clinical trials: proposals for standard definitions and endpoints. *Lancet Oncol* 13: e240-e248
- Griffioen AW, Mans LA, de Graaf AM, Nowak-Sliwinska P, de Hoog CL, de Jong TA, Vyth-Dreese FA, van Beijnum JR, Bex A, Jonasch E (2012) Rapid angiogenesis onset after discontinuation of sunitinib treatment of renal cell carcinoma patients. *Clin Cancer Res* 18: 3961-3971

Kroon BK, de Bruijn R, Prevo W, Horenblas S, Powles T, Bex A (2013) Probability of downsizing primary tumors of renal cell carcinoma by targeted therapies is related to size at presentation. *Urology* 81: 111-115

Tracz A, Mastri M, Lee CR, Pili R, Ebos JM (2014) Modeling spontaneous metastatic renal cell carcinoma (mRCC) in mice following nephrectomy. *Journal of visualized experiments : JoVE*

Appendix

Appendix Figure A1: A) Survival curve comparison of Vehicle-treated control groups from multiple neoadjuvant studies. Expt, Experiment; AVG, Average

Appendix Figure A2: A) Figure 6 from the manuscript (LM2-4^{LUC+} model) sorted by the effect on primary tumor.

A

B

Appendix Figure A3: **A)** Resected tumor weights from LM24-LUC⁺ tumors detailed in Figure 3A-B, and 3D-E following 14 or 7 day treatment with vehicle control or with 60mg/kg/day sunitinib. **B)** Overall postsurgical survival from vehicle-treated control mice treated for 14 or 7 days, as described in Figure 3A-B, and 3D-E.

Thank you for the submission of your manuscript to EMBO Molecular Medicine. We have now heard back from the two Reviewers whom we asked to evaluate your manuscript.

Although the two Reviewers are globally positive, Reviewer 1 has a few remaining concerns that I suggest you address before we proceed with your manuscript. Provided you do so clearly and fully, I am willing to make an editorial decision on your revised, final version. Please provide an additional copy of your revised manuscript with highlighted changes.

Every published paper now includes a 'Synopsis' to further enhance discoverability. Synopses are displayed on the journal webpage and are freely accessible to all readers. They include a short standfirst (to be written by the editor) as well as 2-5 one-sentence bullet points that summarise the paper (to be written by the authors). Please provide the short list of bullet points that summarise the key NEW findings in the passive voice. The bullet points should be designed to be complementary to the abstract - i.e. not repeat the same text. We encourage inclusion of key acronyms and quantitative information. Please attach these in a separate file or send them by email, we will incorporate them accordingly.

I look forward to receiving your revised manuscript as soon as possible, and in any case within two weeks.

***** Reviewer's comments *****

Referee #1 (Remarks):

General Comment:

Overall, the authors have sufficiently addressed the concerns outlined in the initial reviewer assessment and need to be commended for this thorough revision of the original manuscript. Indeed, the changes have mostly helped to clarify the points made and strengthened their conclusions. However, the data interpretation of the rather important and complex new figure 3C and D is somewhat confusing to me and appears to be at odds with the conclusion for panels A-C of the same figure. With this in mind I shall be satisfied with the revised manuscript if this specific issue could be addressed for the benefit of my own understanding and that of the potential readers.

Specific comments

Major:

Fig. 3C

While I am totally happy with the new fig. 3A-B, in fig. 3C the authors depict a grouped analysis of different tumor models according to different antiangiogenic treatments. Referring to this figure the authors state in their general comments to the editor: "there is a clear (and statistically significant) difference in the neoadjuvant efficacy of the TKIs vs the B20 antibody, and strengthens the finding that the anti-primary and anti-metastatic properties for these treatments are not similar across multiple tumor models (Figure 3C)."

Sorry, I am confused by this - is it not the point of grouping the data from different tumor models to arrive at an overriding conclusion regarding the efficacy of a treatment modality that is valid across the different tumor models used? I suppose the black cross in each panel represents the mean value across different models. However, if the results are NOT similar across multiple models (which I agree, the distribution of the data points suggests) then how is the grouped data analysis interpretable? In addition, given that different tumor models are pooled in each group - how can the

results be compared between these disparate groups?

Fig. 3D

There is further issue regarding Fig. 3D. In Figs 3A-C, the authors show that there is a differential response comparing extracellular VEGF inhibitors and TKI. Yet in their Spearman Rank correlation Fig. 3D, they appear to say the opposite, namely, that regardless of the treatment modality, the primary tumor response is predictive of post surgical benefits - am I missing something here?

Minor:

Fig.3C:

The open symbols, and the black crosses are not explained in the figure legend or anywhere else for that matter, please correct.

Fig. 5

As previously requested, could the authors please state the tumor model above the figure panels for added clarity.

Fig. S7D

The outline of the magnified section does not correspond well with the images shown and the lower panel lacks the outlines completely. Please correct.

Referee #2 (Comments on Novelty/Model System):

This paper presents an extraordinary set of mouse models/drug regimens designed to address clinically relevant questions

Referee #2 (Remarks):

The paper has been significantly improved by the suggested revisions. The authors have provided adequate responses to all criticisms, including reasons for not performing some of the suggested studies. This is an outstanding study, presenting a wealth of data with translational robustness and obvious clinical relevance.

Referee #1 (Remarks):

General Comment:

Overall, the authors have sufficiently addressed the concerns outlined in the initial reviewer assessment and need to be commended for this thorough revision of the original manuscript. Indeed, the changes have mostly helped to clarify the points made and strengthened their conclusions.

However, the data interpretation of the rather important and complex new figure 3C and D is somewhat confusing to me and appears to be at odds with the conclusion for panels A-C of the same figure. With this in mind I shall be satisfied with the revised manuscript if this specific issue could be addressed for the benefit of my own understanding and that of the potential readers.

Specific comments

Major:

Fig. 3C

While I am totally happy with the new fig. 3A-B, in fig. 3C the authors depict a grouped analysis of different tumor models according to different antiangiogenic treatments. Referring to this figure the authors state in their general comments to the editor: "there is a clear (and statistically significant) difference in the neoadjuvant efficacy of the TKIs vs the B20 antibody, and strengthens the finding that the anti-primary and anti-metastatic properties for these treatments are not similar across multiple tumor models (Figure 3C)."

Sorry, I am confused by this - is it not the point of grouping the data from different tumor models to arrive at an overriding conclusion regarding the efficacy of a treatment modality that is valid across the different tumor models used? I suppose the black cross in each panel represents the mean value across different models. However, if the results are NOT similar across multiple models (which I agree, the distribution of the data points suggests) then how is the grouped data analysis interpretable? In addition, given that different tumor models are pooled in each group - how can the results be compared between these disparate groups?

We thank the reviewer for this comment and we agree that it is critical this be clear to a reader. The description in our rebuttal may be confusing the message of these results and we would like to restate our analysis and observation.

The purpose of Figure A is to show that TKIs and antibodies produce a significant benefit pre-surgically. Figure B shows that post-surgical survival improvements are only observed with the antibodies, not the TKIs. Figure 3C has the identical message, with the difference that we included data from 2+ models in a grouped analysis. We compared treatments with 2 TKIs (axitinib, sunitinib) and 1 antibody (B.20). Each datapoint shown represents a mouse from a particular tumor model and the value is determined by comparison to the vehicle-treated average for the same model (ie., a sunitinib-treated RENCA animal is compared to the vehicle-treated RENCA average). These comparisons were made for the presurgical tumor burden (either tumor or kidney weight) and the postsurgical survival (median survival). The outcome of this analysis shows again that all 3 drugs produce the same statistical benefit in primary tumor inhibition but only the antibody yielded benefit in significantly improving overall survival. This is shown in Figure 3D in the pre- and post-surgical columns. We have altered the message in the text to explain that the lack of correlation between pre-and post-surgical measurements for the TKIs represents a difference between the antibodies (which, contrast, show consistent correlations in the pre and post-surgical setting). We explained this as 'being not similar' in our response but realize this is a confusing

terminology. We meant to express that this is confirmation of a differential neoadjuvant efficacy between the TKIs and the antibodies, which is similar across tumor models when using this grouped analysis format. We have modified this results section to improve the message. We hope this provides more clarification.

Fig. 3D

There is further issue regarding Fig. 3D. In Figs 3A-C, the authors show that there is a differential response comparing extracellular VEGF inhibitors and TKI. Yet in their Spearman Rank correlation Fig. 3D, they appear to say the opposite, namely, that regardless of the treatment modality, the primary tumor response is predictive of post surgical benefits - am I missing something here?

We agree that this is a confusing concept and we have attempted to make it as clear as possible in the text and in the methods section. This analysis of the pre- and post-surgical data together as individual data-points by spearman rank correlation showed that the *magnitude* of the presurgical response (which represented as a value representing tumor size either larger or smaller than the control average) correlates with the postsurgical survival. This means that a resected tumor much larger than the control average will likely succumb to metastatic disease sooner, regardless of the impact of treatment (i.e., TKI or antibody). This is different than the separate pre- and post-surgical analysis (i.e., Figure 3A and B) in that it considers both outcomes together. In many respects, this result is similar to the analysis shown in Supplemental Figure 1 where we highlight the relationship between tumor size and (at surgery) and eventual survival (after surgery). These results show that the larger the tumor, the sooner the mouse is likely to die. The analysis in Figure 3D shows a similar relationship, however instead this includes the combined pre-/post- neoadjuvant treatment groups and is presented as a relationship to the vehicle treated control groups. In the paper discussion we make the following comment about this in the discussion section:

Already evidence from retrospective studies investigating pre-surgical cytoreductive sunitinib treatment in RCC suggest that parameters of treatment stage (Bex et al, 2011) and primary tumor reduction (Abel et al, 2011) may play a significant role in patient outcomes. In this regard, our results demonstrating that the magnitude of primary tumor response following neoadjuvant therapy correlates with overall survival could support these findings

Here we are referring to the interesting clinical observation that RCC patients that primary tumor size reduction is an independent predictor of improved survival. Our results seem to support this trend in a neoadjuvant model. What is interesting is that this trend may be significant regardless of whether the treatment itself produces a significant benefit compared to control. We hope this clarifies this point.

Minor:

Fig.3C:

The open symbols, and the black crosses are not explained in the figure legend or anywhere else for that matter, please correct.

The open symbols are explained in a sentence in the legend, though we used the word 'hollow'. We will change to open because this is more understandable. We have added a description of the black crosses to the legend as well.

Fig. 5

As previously requested, could the authors please state the tumor model above the figure panels for added clarity.

This has been added.

Fig. S7D

The outline of the magnified section does not correspond well with the images shown and the lower panel lacks the outlines completely. Please correct.

This has been corrected.

Additional Editorial Correspondence

24 September 2014

Your manuscript is almost ready for acceptance but there is still one missing detail that requires fixing. I apologise for not mentioning this in my last decision letter but there is so much data that I overlooked it.

As per our Author Guidelines, the description of all reported data that includes statistical testing must state the name of the statistical test used to generate error bars and P values, the number (n) of independent experiments underlying each data point (not replicate measures of one sample), and the actual P value for each test (not merely 'significant' or ' $P < 0.05$ '). In your manuscript, a few values appear to be missing, namely for figure panels 1G, 2B.E, 3A and 5C. Rather than intervening on the figure panels, I would suggest that you remedy directly in the figure legends by reporting the exact p values in those cases where they cannot be found in the figures.

You can simply send me the amended manuscript by return email, and since you will be modifying it, you might as well remove the highlighted changes and just incorporate them into the next final version (I have checked them and they are OK).

In the meanwhile I would like to take the opportunity to inform you that I have asked a leader in the field to write a Closeup article (our version of News & Views) to highlight your work and which will be published in the same online issue of EMBO Molecular Medicine. I hope this somewhat tempers the frustration for this small additional delay!

Please send me you revised manuscript soonest.

Additional Author Correspondence

25 September 2014

Thank you for your email. Attached are modified figure files with the p-values added. We chose to alter the figures instead of the text because a) adding to the legends may increase the size significantly (i.e., Figure 3A would require each drug mentioned for the panels) and b) we noticed several corrections needed to the figures. For the latter, we noticed some star values that required alteration. The changes made are the following:

1) 4 stars vs 3 stars: In Figure 3D we use a 4 star limit ($p < 0.0001$) but everywhere else we use 3 star limits ($p < 0.001$). We reduced Figure 3D to 3 stars to keep consistent.

2) changes between 1, 2, and 3 stars: We noticed three instances where the star in the graph was incorrect. In Figure 2B the CTX group had 3 stars (it now has 2 stars - $p = 0.002$); in Figure 3A the CT322 group had 2 stars (it now has 3 stars - $p < 0.001$); in figure 5C the sunitinib+LDM CTX/VBL group had 2 stars (it now has 1 star - $p = 0.039$).

3) change from 1 star to 0 star: Perhaps most important, we realized that the Figure 5C 'sunitinib alone' group had 1 star, but is not significant. This is a mistake in the figure only, not in the text. This same data is shown again in Figure S2 (where it has no stars) and the lack of significance is mentioned in the text repeatedly (this was a topic of discussed in our point-by-point response comments). We removed the star and apologize for this oversight.

We went through the rest of the figures and supplemental to ensure there were no more errors. Additionally, we tried to improve the quality of the axis titles to make fonts and sizes more consistent (and legible).

Finally, I have attached two versions of the updated manuscript. One version has the track changes (with new minor changes made to the star references in the legends) and the other version has all the changes accepted.